# KITE: Knowledge-Guided Probabilistic Modeling for Time Series Forecasting with Exogenous Variables

Hanyin Cheng [1]   Jingrong Zhou [1]   Yang Shu [1]   Chenjuan Guo [✉][1]

## Abstract

Probabilistic forecasting with exogenous variables is vital for decision-making but remains underexplored compared to deterministic methods. We propose KITE, a knowledge-guided probabilistic modeling framework designed to bridge this gap by addressing two key bottlenecks: (1) topological disparity in sampling initialization and (2) spurious covariate correlations during the iterative conditional generation process. KITE introduces a *History-Conditional Manifold* to construct an informative source distribution from historical dynamics, effectively anchoring the starting point closer to the target space. Additionally, a *Knowledge-Guided Conditioning* module is developed to regularize variable interactions using statistical priors, suppressing spurious correlations and enhancing the robustness of covariate conditioning. Extensive experiments demonstrate that KITE outperforms state-of-the-art methods in both deterministic and probabilistic forecasting.

 https://github.com/decisionintelligence/KITE

## 1. Introduction

Time series forecasting serves as a pivotal tool for characterizing system dynamics across domains such as economics (Sezer et al., 2020; Huang et al., 2022), traffic (Wu et al., 2024b; 2025f; Pan et al., 2023a; Fang et al., 2021; Yang et al., 2021), energy (Wang et al., 2024a; Guo et al., 2015; Sun et al., 2022), and AIOps (Lin et al., 2024; Campos et al., 2022; Chen et al., 2023c; Pan et al., 2023b; Lin et al., 2025). However, relying solely on historical endogenous data often fails to fully capture complex system dynamics, thereby constraining accuracy. Incorporating covariates such as weather and macroeconomic indicators offers

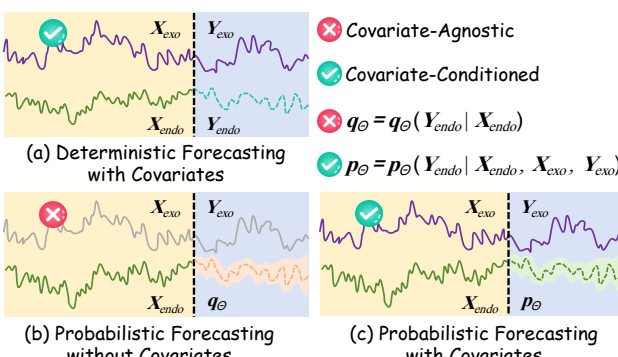

*Figure 1.* We illustrate three distinct prediction paradigms. Specifically, $X_{\mathrm{exo}}$ and $X_{\mathrm{endo}}$ denote historical exogenous and endogenous variables, respectively, while $Y_{\mathrm{exo}}$ and $Y_{\mathrm{endo}}$ represent their counterparts in the future horizon. Furthermore, $q_\theta$ characterizes the covariate-agnostic probabilistic predictive distribution, whereas $p_\theta$ denotes the covariate-conditioned distribution.

critical prospective information that significantly enhances performance (Wang et al., 2024b; Zhou et al., 2025). Consequently, effectively utilizing covariate information has become a prerequisite for achieving high-precision forecasting (Das et al., 2023; Tayal et al., 2024).

While forecasting with covariates yields promising results, existing research remains largely confined to deterministic point forecasting (Arango et al., 2025; Ekambaram et al., 2024), thereby neglecting the inherent stochasticity and uncertainty characterizing future environments, as illustrated in Figure 1(a). Real-world systems fluctuate constantly due to complex factors. Relying on deterministic estimates creates misleading confidence, leading to accumulated errors and risky decisions (Wu et al., 2025c; Rasul et al., 2021a; Tashiro et al., 2021). This limitation severely hinders model performance in uncertain environments. To close this gap, we aim to realize probabilistic forecasting with covariates, a paradigm depicted in Figure 1(c). However, this faces the following challenges:

First, there is a fundamental **inefficiency in mapping context-agnostic priors to covariate-conditioned distributions**. Existing probabilistic models (Hu et al., 2025; Liu et al., 2025; Rishi et al., 2025) typically initialize sampling from a simple, context-agnostic prior (e.g., Gaussian

[1]East China Normal University, Shanghai, China. Correspondence to: Chenjuan Guo <cjguo@dase.ecnu.edu.cn>.

*Proceedings of the 43rd International Conference on Machine Learning*, Seoul, South Korea. PMLR 306, 2026. Copyright 2026 by the author(s).

noise (Goodman, 1963)), resulting in a vast topological disparity between the context-agnostic source distribution of the generative model and the complex target distribution of the ground truth. This discrepancy is particularly pronounced in forecasting with covariates, as the covariate-conditioned distributions are often highly localized and exhibit intricate structures within a high-dimensional space. Such context-agnostic initialization compels the model to reconstruct complex temporal dependencies and dynamic covariate interactions entirely from scratch. Consequently, the induced transport trajectories become complex and non-linear, which significantly increases sampling latency and compromises predictive fidelity. Although certain models, such as TSFlow (Kollovieh et al., 2025), attempt to introduce historically correlated priors, they often rely on fixed, non-learnable rules, leading to a representational bottleneck that limits their expressive capacity.

Second, **the probabilistic generation process exhibits vulnerability to spurious covariate correlations**. Existing covariate modeling methods (Wang et al., 2024b; Arango et al., 2025) rely heavily on purely data-driven learning to implicitly infer interactions among exogenous variables. However, this mechanism often causes models to overfit static spurious correlations within the training data rather than capturing true system dynamics. When such a mechanism is applied to iterative probabilistic generation, this vulnerability is magnified during iterative sampling, as flawed correlations cause errors to propagate and amplify along the transport trajectory. Consequently, when encountering unforeseen out-of-distribution (OOD) relational shifts (Hsu et al., 2020), these methods lack sufficient robustness, ultimately compromising the fidelity of the predictive probability space. Although some correlation modeling studies (Zhao et al., 2023; Cao et al., 2020) incorporate statistical knowledge (Pearson et al., 2004) to bolster robustness, their reliance on naive addition fails to fundamentally resolve the structural bottlenecks inherent in architectures devoid of effective knowledge guidance.

To address these challenges, in this study, we propose **KITE**, a **K**nowledge-Gu**I**ded Probabilistic Modeling for **T**ime Series Forecasting with **E**xogenous Variables. Our method utilizes the advanced Flow Matching framework for generative probabilistic modeling, incorporating covariate information as generation conditions to better regulate variable dependencies. First, to address the limitations caused by sampling initialization with simple uninformative priors, We propose the *History-Conditional Manifold* to serve as the source distribution. Conditioned on historical information, this module dynamically constructs a manifold that closely approximates the target distribution. This effectively bridges the gap between the source and target distributions before the generative flow process even begins, thereby alleviating the burden of learning complex, covariate-aware

structured distributions. Second, to mitigate the capture of spurious covariate correlations and thereby achieve a robust probabilistic generation process, we propose *Knowledge-Guided Conditioning*. In this module, we innovatively integrate statistical knowledge into the attention mechanism for capturing variable dependencies. Specifically, we adopt a bilinear interaction strategy to dynamically modulate the query projection subspace conditioned on the strength of prior structural signals. By doing so, the feature extraction process for attention score calculation adapts to shifts in the statistical environment, effectively enhancing the robustness. Furthermore, by incorporating classifier-free guidance, our method is capable of performing both covariate-agnostic and covariate-conditioned generation simultaneously. This mechanism allows us to explicitly control the degree of covariate influence on the generation process, thereby facilitating flexible adaptation to diverse real-world scenarios. Our contributions are summarized as follows:

- We propose KITE, an end-to-end framework for probabilistic time series forecasting with exogenous variables. Built upon flow matching, KITE treats exogenous variables as conditions to enable more controllable and robust modeling of variable interactions.

- To overcome the limitations of sampling from simple priors, we propose the *History-Conditional Manifold* to dynamically construct a history-aware source distribution, thereby bridging the gap between the source and target distributions prior to the generative flow.

- For the robustness of variable modeling, we design the *Knowledge-Guided Conditioning* to explicitly integrate statistical Knowledge into the attention mechanism, dynamically modulating the projection space to adapt to non-stationary statistical environments.

- Comprehensive experiments on both deterministic and probabilistic forecasting demonstrate that KITE consistently outperforms state-of-the-art baselines. Our code are available at https://github.com/decisionintelligence/KITE.

## 2. Related Works

### 2.1. Time Series Forecasting with Exogenous Variables

The integration of exogenous variables into time series forecasting has its roots in classical statistics, where models such as ARIMAX (Williams, 2001) and SARIMAX (Vagropoulos et al., 2016) pioneered the use of external factors to refine predictive precision. Modern deep learning frameworks have further sophisticated this paradigm: CrossLinear (Zhou et al., 2025) implements cross-correlation embeddings to map dependencies between historical endogenous and ex-

ogenous series. While NBEATSx (Olivares et al., 2023) introduces specialized architectural branches for multi-period exogenous inputs, TiDE (Das et al., 2023) utilizes a dense MLP structure to fuse static and future covariates with endogenous features via concatenation. Attention-based mechanisms have also proven pivotal; for instance, the Temporal Fusion Transformer (TFT) (Lim et al., 2021) employs attention to synthesize historical and concurrent exogenous information. Additionally, TimeXer (Wang et al., 2024b) facilitates flexible covariate integration through patch-wise embeddings, bypassing the need for rigid temporal alignment. Alternatively, ExoTST (Tayal et al., 2024) addresses challenges like time lags and data sparsity by leveraging a combination of cross-attention and cross-temporal fusion. Despite the notable performance gains achieved by these methods (Wang et al., 2024b; Li et al., 2026c), current research remains largely confined to the deterministic modeling, overlooking the critical role of uncertainty quantification in understanding system evolution driven by complex covariates in the future or the past.

### 2.2. Probabilistic Modeling in Time Series Forecasting

Probabilistic forecasting aims to characterize the future predictive distribution. Early deep learning methodologies, such as DeepAR (Salinas et al., 2020), primarily utilized RNNs for temporal transitions, while frameworks like DeepState (Rangapuram et al., 2018) and DSSMF (Li et al., 2019) integrated state-space models to refine precision. More recently, attention-driven models, including MANF (Feng et al., 2024) and ProTran (Tang & Matteson, 2021), have further enhanced long-range dependency modeling. The field has lately shifted toward generative paradigms. Diffusion-based models, such as TimeGrad (Rasul et al., 2021a), TSDiff (Kollovieh et al., 2023), and CSDI (Tashiro et al., 2021), treat forecasting as a score-based denoising process. Within this landscape, TSFlow (Kollovieh et al., 2025) introduces a conditional flow matching framework that leverages Gaussian Process priors to align the generative source with the data's temporal structure. Alternatively, $K^2$VAE (Wu et al., 2025c) linearizes nonlinear dynamics via Koopman Theory and employs a KalmanNet to iteratively refine predictions. While certain methods incorporate multivariate dependencies (Rasul et al., 2021a; Tashiro et al., 2021), they are not natively designed for covariate-conditioned forecasting, particularly in scenarios where future exogenous information is explicitly known.

## 3. Preliminaries

### 3.1. Flow Matching

Our approach utilizes a generative framework to transport samples $z_0$ from a prior distribution $p_0$ to data samples $z_1$ in the target distribution $p_1$, conditioned on $c$. This trans-

formation is defined by an Ordinary Differential Equation, $dz_s = v_\theta(z_s, s, c)ds$, driven by a velocity field $v_\theta$ with learnable parameters $\theta$, where $s \in [0, 1]$ denotes the integration time. Following the formulation of Flow Matching (Lipman et al., 2022; Albergo & Vanden-Eijnden, 2023), we construct the intermediate state $z_s$ via linear interpolation:

$$z_s = s \cdot z_1 + (1 - s) \cdot z_0 \tag{1}$$

Consequently, the target velocity field corresponds to the time derivative of the path:

$$\hat{v}_s = z_1 - z_0 \tag{2}$$

The neural network $v_\theta(z_s, s, c)$ is optimized to approximate this vector field by minimizing the difference between the predicted velocity and the analytical target $\hat{v}_s$, effectively learning a straight-line trajectory from the source to the target distribution.

Flow Matching enables the transformation of samples from an arbitrary source distribution $p_0$ to target data $z_1$. However, prior work (Esser et al., 2024; Polyak et al., 2024) typically initializes $z_0$ from Gaussian noise $\mathcal{N}(0, 1)$, which may have an inherent bias relative to the target distribution.

### 3.2. Problem Statement

We address the task of Probabilistic Time Series Forecasting with Exogenous Variables. Given the history of $N$ endogenous variables $\boldsymbol{X}_{\text{endo}} \in \mathbb{R}^{N \times T}$ over $T$ time steps, and $D$ exogenous covariates available for both the past ($\boldsymbol{X}_{\text{exo}} \in \mathbb{R}^{D \times T}$) and the future horizon of length $F$ ($\boldsymbol{Y}_{\text{exo}} \in \mathbb{R}^{D \times F}$), our goal is to model the future endogenous values $\boldsymbol{Y}_{\text{endo}} \in \mathbb{R}^{N \times F}$. Unlike deterministic forecasting which yields a single point estimate, we aim to estimate the conditional probability density:

$$p_\theta(\boldsymbol{Y}_{\text{endo}} \mid \boldsymbol{X}_{\text{endo}}, \boldsymbol{X}_{\text{exo}}, \boldsymbol{Y}_{\text{exo}}), \tag{3}$$

modeling the distribution of future trajectories given the available context.

## 4. Methodology

### 4.1. KITE Architecture

As illustrated in Figure 2, KITE employs Flow Matching as its generative backbone and utilizes exogenous covariates as conditional inputs to achieve effective and controllable distribution estimation.

The framework comprises three core components:

(1) *History-Conditional Manifold* (HCM): Taking historical endogenous data as input, this module constructs a structured source distribution. Unlike standard Gaussian noise, HCM is highly informative and offers a source distribution

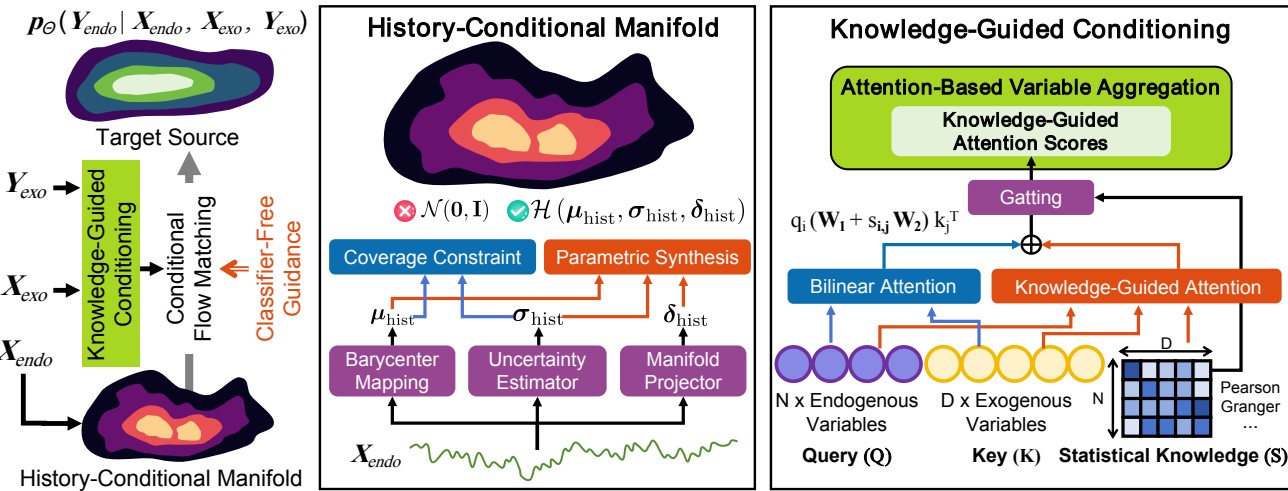

*Figure 2.* The architecture of KITE. Here, $\boldsymbol{X}_{\text{exo}}$ and $\boldsymbol{X}_{\text{endo}}$ denote historical exogenous and endogenous variables, respectively, $\boldsymbol{Y}_{\text{exo}}$ represent future exogenous variables, and $s_{i,j}$ denotes the strength of the statistical knowledge between the $i$-th endogenous variable and the $j$-th exogenous variable.

positioned topologically closer to the target distribution, thereby improving transport efficiency.

(2) *Knowledge-Guided Conditioning* (KGC): This module leverages a bilinear attention mechanism to capture the dependencies between endogenous and exogenous variables, injecting them as conditions for the Flow Matching process. To enhance the robustness of dependency modeling, the module further integrates statistical knowledge to mitigate spurious correlations.

(3) *Classifier-Free Guidance* (CFG): During both training and inference, this strategy effectively modulates the interplay between covariate-agnostic and covariate-conditioned generation, enhancing the controllability and precision.

Overall, we utilize historical endogenous data to construct the HCM, which serves as the source distribution for Flow Matching. During the sampling process, the KGC module extracts covariate correlations to condition the generation. Furthermore, both training and inference phases are governed by CFG.

### 4.2. History-Conditional Manifold

Unlike prior works (Kollovieh et al., 2025) that rely on non-learnable, rule-based generation of context-aware source distributions, we introduce the adaptively learnable *History-Conditional Manifold* to replace the standard source typically employed in Flow Matching. To facilitate more efficient transport trajectories, we construct an informative source distribution $\mathcal{H}(\mu_{\text{hist}}, \sigma_{\text{hist}}, \delta_{\text{hist}})$, where source samples are synthesized as $Y_0 = \mu_{\text{hist}} + \sigma_{\text{hist}}\delta_{\text{hist}}$. Specifically, $\mu_{\text{hist}}$ captures the history-dependent central tendency, $\sigma_{\text{hist}}$ controls the adaptive uncertainty scale, and $\delta_{\text{hist}}$ provides structured perturbations. This construction leverages tem-

poral dependencies to bridge the topological gap between the source and target distributions. We further substantiate the advantages of the *History-Conditional Manifold* in Section 4.5, where we analyze how the history-conditioned source contributes to a smaller matching target scale and a more regular transport path.

**Barycenter Mapping.** This module defines the first-order moment of the source distribution. It projects the historical context into a central position within the proximity of the target prediction space:

$$\mu_{\text{hist}} = f_\phi(\boldsymbol{X}_{\text{endo}}) \in \mathbb{R}^{N \times F}, \tag{4}$$

where $f\phi$ denotes a linear mapping or a Multi-Layer Perceptron (MLP) parameterized by $\phi$.

**Uncertainty Estimator.** This module is responsible for estimating the scalar scale of the distribution to achieve heteroscedastic modeling, allowing the model to adaptively adjust the coverage of the source distribution based on the complexity of the historical data:

$$\sigma_{\text{hist}} = \text{Softplus}(g_\psi(\boldsymbol{X}_{\text{endo}})) + \sigma_{\min}, \tag{5}$$

where $g_\psi$ is a MLP that outputs a scalar. The Softplus function ensures that the standard deviation remains positive ($\sigma_{\text{hist}} > 0$), while the term $\sigma_{\min}$ is incorporated to guarantee a baseline exploration width for the distribution.

**Manifold Projector.** By employing a learnable Manifold Basis to map random noise into a low-rank feature space, the module induces an anisotropic prior that aligns with the geometric structure of the target data. To prevent overfitting to this low-rank manifold and capture subtle perturbations, we also incorporate isotropic noise:

$$\delta_{\text{hist}} = \alpha \cdot \frac{\boldsymbol{Mz}}{||\boldsymbol{Mz}||} + (1 - \alpha) \cdot \epsilon \in \mathbb{R}^{N \times F}, \tag{6}$$

where $M \in \mathbb{R}^{(N \times F) \times r}$ denotes the learnable Manifold Basis with rank $r$; both the latent vector $z \in \mathbb{R}^r$ and the isotropic noise $\epsilon \in \mathbb{R}^{N \times F}$ are sampled from a standard normal distribution; and the mixing weight $\alpha$ is a learnable parameter to balance the proportion between structured and isotropic noise.

**Coverage Constraint.** To prevent the learnable variance from collapsing towards zero and to force the support of the source distribution to encompass the ground-truth target values, we introduce the following objective:

$$\mathcal{L}_{CC} = \frac{1}{2} \left( \log \sigma_{\text{hist}}^2 + \frac{||Y_{\text{endo}} - \text{detach}(\mu_{\text{hist}})||^2}{\sigma_{\text{hist}}^2} \right), \quad (7)$$

where $\text{detach}(\cdot)$ denotes the stop-gradient operation, ensuring that this constraint does not interfere with the geometric optimization of the barycenter.

**Parametric Synthesis.** For the $s$-th sampling step in the Flow Matching process, the latent state $Y_s$ is defined by the probability path:

$$Y_s = s \cdot Y_{\text{endo}} + (1 - s) \cdot Y_0, \quad (8)$$

where the source distribution is parameterized as following:

$$Y_0 = \mu_{\text{hist}} + \sigma_{\text{hist}} \delta_{\text{hist}}. \quad (9)$$

This synthesis ensures that the source distribution $Y_0$ is not a standard isotropic Gaussian, but a knowledge-informed prior that is topologically and statistically aligned with the target manifold $Y_{\text{endo}}$, thereby significantly rectifying the transport trajectory.

### 4.3. Knowledge-Guided Conditioning

Our modeling approach is conditioned on future and past covariates, achieving the conditioning process by capturing the correlations between endogenous and exogenous variables. To enhance the capability and robustness in capturing the underlying mechanisms of complex systems, we incorporate statistical knowledge (e.g., Pearson correlation (Pearson, 1895), Granger causality (Granger, 1969)), which is explicitly extracted from the input context, as a guidance.

Distinct from conventional methods that directly concatenate prior weights at the input stage, we proposes a modulation strategy based on a bilinear attention mechanism. We project the query vectors into two distinct subspaces and perform dynamic interpolation between the interactions of these subspaces using the statistical knowledge $S = \{s_{ij} \mid 1 \le i \le N, 1 \le j \le D\} \in \mathbb{R}^{N \times D}$. Here, $s_{ij}$ denotes the strength of the statistical knowledge between the $i$-th endogenous variable and the $j$-th exogenous variable. The core formulation can be expressed as:

$$\text{Attn}(q_i, k_j) = q_i(\mathbf{W}_1 + s_{ij} \cdot \mathbf{W}_2)k_j^\top, \quad (10)$$

where, $q_i$ and $k_j$ represent the query and key vectors for the $i$-th endogenous and $j$-th exogenous elements, respectively, while $s_{ij}$ denotes their statistical prior. By interpolating between a data-driven subspace ($W_1$) and a prior-injected subspace ($W_2$), this mechanism utilizes this prior knowledge to dynamically rectify the dependency structure, effectively grounding the attention mechanism in structural priors.

In practice, we implement the two interaction subspaces in Eq. 10 using two factorized query-key projection branches, without explicitly forming the full bilinear matrices. We inject the historical and future covariates sequentially using the same methodology. In scenarios where future covariates are unavailable, the corresponding injection step can be omitted. For demonstration, we illustrate the process using only historical covariates as follows:

$$\mathbf{A}_b = (\mathbf{Y}_s \mathbf{W}_{1,q})(\mathbf{X}_{\text{exo}} \mathbf{W}_{1,k})^\top / \sqrt{d}, \quad (11)$$

$$\mathbf{A}_g = (\mathbf{Y}_s \mathbf{W}_{2,q})(\mathbf{X}_{\text{exo}} \mathbf{W}_{2,k})^\top / \sqrt{d}, \quad (12)$$

where $\mathbf{W}_{1,q}, \mathbf{W}_{2,q} \in \mathbb{R}^{F \times d}$, $\mathbf{W}_{1,k}, \mathbf{W}_{2,k} \in \mathbb{R}^{T \times d}$ are the weight matrices employed to derive the query and key vectors, respectively. The final attention matrix $\mathbf{A}$ is then formulated as follows:

$$\mathbf{A} = \text{Softmax} \left( \mathbf{A}_b + \tilde{\mathbf{S}} \odot \mathbf{A}_g + \log(\tilde{\mathbf{S}} + \delta) \right), \quad (13)$$

where $\delta$ is a fixed small constant for numerical stability; $\tilde{\mathbf{S}} = \text{Norm}(|\mathbf{S}|) \in [0, 1]^{N \times D}$ denotes the normalized non-negative statistical guidance matrix, and $\text{Norm}(\cdot)$ represents normalization that rescales the statistical prior magnitude into $[0, 1]$. We further incorporate a gating mechanism through the logarithmic term $\log(\tilde{\mathbf{S}} + \delta)$, which ensures that the attention score $\mathbf{A}_k$ is strongly suppresses when the statistical knowledge is zero. The final conditional injection is formulated as follows:

$$\tilde{\mathbf{Y}}_s = \mathbf{A} \mathbf{X}_{\text{exo}} \mathbf{W}_v + \mathbf{Y}_s \in \mathbb{R}^{N \times F}, \mathbf{W}_v \in \mathbb{R}^{T \times F}. \quad (14)$$

### 4.4. Pipeline with Classifier-Free Guidance

Widely adopted to enhance generation quality, Classifier-Free Guidance (CFG) (Ho & Salimans, 2022) employs a unified training strategy for conditional generative models. This mechanism has proven to be a pivotal component in advanced image generation frameworks (Esser et al., 2024; Chen et al., 2023a). Furthermore, we incorporate CFG into our framework to modulate the interplay between Covariate-Agnostic and Covariate-Conditioned generation.

**Training strategy.** By intermittently omitting the input condition $c = \{X_{\text{exo}}, Y_{\text{exo}}\}$ with probability $p_{\text{con}}$, the model simultaneously learns both covariate-conditioned and covariate-agnostic distributions. Specifically, during each training iteration, we define a null condition $\emptyset$ and generate

a masked condition $\tilde{c}$ as follows:

$$\tilde{c} = \begin{cases} \emptyset & \text{with probability } p_{\text{con}} \\ c & \text{with probability } 1 - p_{\text{con}} \end{cases} \quad (15)$$

The velocity field network $v_\theta$ is then optimized using the following Flow Matching objective:

$$\mathcal{L}_{FM} = |v_\theta(\boldsymbol{Y}_s, s, \tilde{c}) - (\boldsymbol{Y}_{\text{endo}} - \boldsymbol{Y}_0)| \quad (16)$$

By training with this stochastic masking, the model $v_\theta$ learns to estimate the vector field with or without covariate.

**Inference stage.** During the inference phase, the final guided velocity field $\hat{\boldsymbol{v}}_s$ is extrapolated by linearly combining these two learned modes:

$$\hat{\boldsymbol{v}}_s(\boldsymbol{Y}_s, c) = (1 + \gamma) \cdot \boldsymbol{v}_\theta(\boldsymbol{Y}_s, s, c) - \gamma \cdot \boldsymbol{v}_\theta(\boldsymbol{Y}_s, s, \emptyset), \quad (17)$$

where $\gamma \geq 0$ is a hyperparameter for guidance scale. This formulation enables the model to steer the generative trajectory away from the generic covariate-agnostic path and towards a direction that is more strictly governed by the specific conditional drivers, thereby enhancing the precision and robustness of the forecast. The complete sampling procedure is summarized in Algorithm 1.

### 4.5. Theoretical Analysis

We theoretically analyze the advantages of initializing flow matching from *History-Conditional Manifold* by investigating its **target scale** and **path regularity**.

#### 4.5.1. TARGET SCALE

**proposition 1** (Target Scale). Let $Y$ be the target, $h := X_{\text{endo}}$ the endogenous context, and $c$ the covariates. Let $Y_0^H = \mu(h) + \sigma(h)\delta$ denote the HCM source, where $\mu(h)$ is the history-dependent barycenter, $\sigma(h)$ is the history-dependent scale, and $\delta$ is the HCM perturbation. Let $Y_0^G = \xi \sim \mathcal{N}(0, I)$ denote the standard Gaussian source. Define their expected $L_1$ noise magnitudes as $C_H := \mathbb{E}[\|\delta\|_1]$ and $C_G := \mathbb{E}[\|\xi\|_1]$. If the conditional residual reduction in centering error $\mathbb{E}[\|Y\|_1 \mid h, c] - \mathbb{E}[\|Y - \mu(h)\|_1 \mid h, c]$ provided by $\mu(h)$ exceeds the combined expected noise magnitudes $\sigma(h)C_H + C_G$, then HCM yields a smaller conditional matching target than the standard Gaussian source:

$$\mathbb{E}[\|Y - Y_0^H\|_1 \mid h, c] < \mathbb{E}[\|Y - Y_0^G\|_1 \mid h, c]. \quad (18)$$

**Remark.** The assumption is natural in forecasting, where future values typically concentrate around a history-dependent local center rather than the origin of $\mathcal{N}(0, I)$, making the conditional residual reduction more likely to be sufficiently large. Meanwhile, Eq. (7) calibrates $\sigma(h)$, and the normalized manifold perturbation keeps the added noise controlled. Thus, HCM reduces the regression scale, stabilizing gradients and alleviating the empirical optimization burden.

#### 4.5.2. PATH REGULARITY

**Proposition 2** (Path Regularity). Let $Y_s^H = sY + (1-s)Y_0^H$ and $Y_s^G = sY + (1-s)Y_0^G$ denote the straight-line matching paths under the HCM source $Y_0^H = \mu(h) + \sigma(h)\delta$ and the standard Gaussian source $Y_0^G = \xi \sim \mathcal{N}(0, I)$, respectively. Define their path-tube radii as $R_H := \sup_{s \in [0,1]} \|Y_s^H - \mu(h)\|_1$ and $R_G := \sup_{s \in [0,1]} \|Y_s^G\|_1$. Suppose the vector field satisfies a local regularity condition bounding its Jacobian norm by the spatial radius $r$ around any center $m$:

$$\sup_{\substack{s \in [0,1] \\ \|y - m\|_1 \leq r}} \|\nabla_y v_\theta(y, s, c)\|_{1 \to 1} \leq a + br \quad (a \geq 0, b > 0). \quad (19)$$

Under this assumption, a path tube of radius $R$ admits the Jacobian upper bound $a + bR$, which quantifies how strongly the vector field can amplify local perturbations along the path. Accordingly, define the corresponding path-local Jacobian budgets as $\mathcal{B}_H := a + bR_H$ and $\mathcal{B}_G := a + bR_G$. If the condition of Proposition 4.1 holds, then HCM yields a smaller expected path-local Jacobian budget than the standard Gaussian source:

$$\mathbb{E}[\mathcal{B}_H \mid h, c] < \mathbb{E}[\mathcal{B}_G \mid h, c]. \quad (20)$$

**Remark.** Proposition 4.2 assumes a local, rather than global, regularity condition. This is reasonable for smooth neural vector fields, since locally bounded Hessians on a fixed tube imply such a bound. Under this assumption, Proposition 4.2 shows that HCM keeps the straight-line matching path in a narrower tube around $\mu(h)$ than the standard source when Proposition 4.1 holds. This yields a smaller path-local Jacobian budget and supports weaker local error amplification under standard ODE stability arguments.

We provide the formal proofs for **Proposition 4.1 – 4.2** in Appendix A.

## 5. Experiments

In this section, we provide empirical results to show the strong performance of KITE against state-of-art baselines on both deterministic and probabilistic forecasting with exogenous variables.

### 5.1. Experimental Setup

#### 5.1.1. DATASETS

**Deterministic forecasting.** We evaluate KITE on 12 real-world datasets characterized by the presence of exogenous variables, comprising 5 Electricity Price Forecasting (Lago et al., 2021; Wang et al., 2024b) datasets and 7 other sourced datasets. In these datasets, future exogenous variables are either explicitly known or provided with high precision.

*Table 1.* Statistics of all datasets for deterministic and probabilistic forecasting. #Num denotes the number of exogenous variables. Ex. and En. are abbreviations for the exogenous variable and endogenous variable, respectively.

| Dataset | #Num | Ex. Descriptions | En. Descriptions | Sampling Frequency | Lengths | Split |
|---|---|---|---|---|---|---|
| NP | 2 | Grid Load, Wind Power | Nord Pool Electricity Price | 1 Hour | 52,416 | 7:1:2 |
| PJM | 2 | System Load, SyZonal COMED load | Pennsylvania-New Jersey-MarylandElectricity Price | 1 Hour | 52416 | 7:1:2 |
| BE | 2 | Generation, System Load | Belgium's Electricity Price | 1 Hour | 52,416 | 7:1:2 |
| FR | 2 | Generation, System Load | France's Electricity Price | 1 Hour | 52,416 | 7:1:2 |
| DE | 2 | Wind power, Ampirion zonal load | German's Electricity Price | 1 Hour | 52,416 | 7:1:2 |
| Energy | 5 | Battery, Geothermal, Hydroelectric, Solar, Wind | Thermoelectric | 1 Hour | 13,064 | 7:1:2 |
| Colbun | 2 | Precipitation, Tributary Inflow | Water Level | 1 Day | 2,958 | 7:1:2 |
| Rapel | 2 | Precipitation, Tributary Inflow | Water Level | 1 Day | 3,366 | 7:1:2 |
| Sdwpfh | 6 | Climate Feature | Active Power | 1 Hour | 1,4641 | 7:1:2 |
| Sdwpfm | 6 | Climate Feature | Active Power | 30 Minutes | 2,9281 | 7:1:2 |
| ETTh | 6 | Power Load Feature | Oil Temperature | 1 Hour | 14,400 | 6:2:2 |
| ETTm | 6 | Power Load Feature | Oil Temperature | 15 Minutes | 57,600 | 6:2:2 |
| Weather | 20 | Climate Feature | CO2-Concentration | 10 Minutes | 52,696 | 7:1:2 |
| Exchange | 7 | Exchange Rate | Exchange Rate | 1 Day | 7,588 | 7:1:2 |
| Electricity | 320 | Electricity Consumption | Electricity Consumption | 1 Hour | 26,304 | 7:1:2 |
| Traffic | 861 | Road Occupancy Rates | Road Occupancy Rates | 1 Hour | 17,544 | 7:1:2 |

**Probabilistic Forecasting.** We also conduct extensive experiments on 8 widely recognized multivariate time series datasets based on ProbTS (Zhang et al., 2024a), a comprehensive benchmark for probabilistic forecasting tasks. Notably, we follow existing works (Wang et al., 2024b; Lim et al., 2021) to conduct vanilla forecasting with exogenous variables. Specifically, we partition the multivariate data by designating the last dimension as the target endogenous series, while treating the remaining dimensions as the associated exogenous variables.

Table 1 summarizes the statistics of all datasets for deterministic and probabilistic forecasting.

### 5.1.2. BASELINES

**Deterministic forecasting.** We compare KITE against 10 competitive baselines. These are categorized into: 1) architectures with native support for future exogenous variables, including TimeXer (Wang et al., 2024b), TFT (Lim et al., 2021), and TiDE (Das et al., 2023); and 2) models originally designed without such support, namely DUET (Qiu et al., 2025c), CrossLinear (Zhou et al., 2025), Amplifier (Fei et al., 2025), TimeKAN (Huang et al., 2025), xPatch (Stit-syuk & Choi, 2025), PatchTST (Nie et al., 2023b), and DLinear (Zeng et al., 2023). For the latter group, we implement a modification to incorporate future covariates using an MLP-based fusion strategy, as described in Algorithm 4.

**Probabilistic Forecasting.** We select 10 strong baselines, including 5 univariate models: PatchTST (Nie et al., 2023a), TSDiff (Kollovieh et al., 2023), Koopa (Liu et al., 2023), TSFlow (Kollovieh et al., 2025), and $K^2$VAE, as well as 5 multivariate models: iTransformer (Liu et al., 2024b), GRU NVP, Trans MAF (Rasul et al., 2021c), TimeGrad (Rasul et al., 2021a), and CSDI (Tashiro et al., 2021). The point forecasting models are equipped with gaussian heads to predict the distributions.

### 5.1.3. IMPLEMENTATION DETAILS

To keep consistent with previous works, we adopt Mean Squared Error (MSE) and Mean Absolute Error (MAE) as evaluation metrics for Deterministic forecasting and Continuous Ranked Probability Score (CRPS) for Probabilistic forecasting. Following the settings in TFB (Qiu et al., 2024), we do not apply the "Drop Last" trick to ensure a fair comparison. All experiments are conducted using PyTorch (Paszke et al., 2019) in Python 3.8 and executed on an NVIDIA Tesla-A800 GPU. To ensure experimental consistency, we select Pearson correlation coefficients (Pearson, 1895) as the foundational statistical knowledge for *Knowledge-Guided Conditioning*.

Comprehensive details on datasets, baselines, metrics, and implementation can be found in Appendix D (specifically Sections D.1–D.4).

### 5.2. Main Results

Comprehensive deterministic and probabilistic forecasting results with exogenous variables are listed in Table 2, Table 3 and Table 4 with the best in red and the second in blue. We have the following observations:

1) As shown in Table 2, KITE outperforms state-of-the-art deterministic baselines, demonstrating notable improvements in predictive performance. In the presence of future exogenous variables, KITE achieves a *16.1%* reduction in MSE compared to CrossLinear, the second-best baseline in terms of MSE, and a *9.2%* reduction in MAE compared to TFT, the second-best baseline in terms of MAE. Full results are reported in Table 8. This compellingly demonstrates the advantages of probabilistic modeling in forecasting tasks with exogenous variables.

2) The deterministic forecasting results without future exogenous variables are reported in Table 3 for average results and Table 9 for full results. In most cases, KITE outper-

*Table 2.* Average results on deterministic forecasting with historical and future exogenous variables across 12 real-world datasets, where the inputs are $X_{endo}$, $X_{exo}$, and $Y_{exo}$. **Red**: the best, Blue: the 2nd best. Full results with $Y_{exo}$ are reported in Table 8.

| Models | KITE (ours) | | TimeXer | | TFT | | TiDE | | DUET | | CrossLinear | | Amplifier | | TimeKAN | | xPatch | | PatchTST | | DLinear | |
|---|---|---|---|---|---|---|---|---|---|---|---|---|---|---|---|---|---|---|---|---|---|---|
| Metrics | MSE | MAE | MSE | MAE | MSE | MAE | MSE | MAE | MSE | MAE | MSE | MAE | MSE | MAE | MSE | MAE | MSE | MAE | MSE | MAE | MSE | MAE |
| NP | **0.325** | **0.323** | 0.418 | 0.370 | 0.379 | 0.375 | 0.443 | 0.399 | 0.411 | 0.407 | 0.370 | 0.387 | 0.419 | 0.418 | 0.406 | 0.419 | 0.378 | 0.370 | 0.390 | 0.396 | 0.442 | 0.444 |
| PJM | **0.096** | **0.178** | 0.108 | 0.199 | 0.114 | 0.207 | 0.141 | 0.246 | 0.102 | 0.197 | 0.112 | 0.223 | 0.137 | 0.246 | 0.139 | 0.263 | 0.104 | 0.194 | 0.133 | 0.259 | 0.130 | 0.243 |
| BE | **0.427** | **0.286** | 0.452 | 0.290 | 0.454 | 0.291 | 0.498 | 0.325 | 0.514 | 0.354 | 0.479 | 0.338 | 0.558 | 0.413 | 0.548 | 0.407 | 0.495 | 0.333 | 0.577 | 0.432 | 0.526 | 0.376 |
| FR | **0.387** | **0.225** | 0.427 | 0.240 | 0.504 | 0.257 | 0.485 | 0.281 | 0.495 | 0.327 | 0.482 | 0.298 | 0.553 | 0.408 | 0.547 | 0.374 | 0.490 | 0.293 | 0.588 | 0.410 | 0.497 | 0.314 |
| DE | **0.350** | **0.370** | 0.475 | 0.418 | 0.489 | 0.446 | 0.498 | 0.447 | 0.482 | 0.430 | 0.485 | 0.452 | 0.473 | 0.441 | 0.473 | 0.446 | 0.474 | 0.433 | 0.501 | 0.455 | 0.498 | 0.457 |
| Energy | **0.111** | **0.257** | 0.163 | 0.315 | 0.130 | 0.283 | 0.152 | 0.301 | 0.202 | 0.367 | 0.239 | 0.401 | 0.233 | 0.389 | 0.218 | 0.381 | 0.148 | 0.303 | 0.226 | 0.377 | 0.174 | 0.323 |
| Sdwpfm1 | **0.417** | **0.451** | 0.702 | 0.609 | 0.481 | 0.475 | 0.483 | 0.507 | 0.599 | 0.571 | 0.426 | 0.502 | 0.428 | 0.490 | 0.447 | 0.534 | 0.630 | 0.568 | 0.435 | 0.502 | 0.449 | 0.531 |
| Sdwpfm2 | **0.475** | **0.483** | 0.802 | 0.653 | 0.476 | 0.489 | 0.486 | 0.516 | 0.514 | 0.490 | 0.533 | 0.574 | 0.490 | 0.512 | 0.497 | 0.564 | 0.516 | 0.495 | 0.510 | 0.547 | 0.524 | 0.580 |
| Sdwpfh1 | **0.441** | **0.491** | 0.746 | 0.643 | 0.479 | 0.492 | 0.453 | 0.508 | 0.539 | 0.516 | 0.557 | 0.593 | 0.536 | 0.598 | 0.577 | 0.638 | 0.498 | 0.505 | 0.468 | 0.526 | 0.579 | 0.624 |
| Sdwpfh2 | **0.500** | 0.526 | 0.891 | 0.719 | 0.566 | **0.521** | 0.599 | 0.584 | 0.647 | 0.566 | 0.538 | 0.575 | 0.521 | 0.581 | 0.646 | 0.671 | 0.610 | 0.551 | 0.549 | 0.580 | 0.669 | 0.664 |
| Colbun | **0.088** | **0.172** | 0.144 | 0.235 | 0.237 | 0.297 | 0.164 | 0.227 | 0.198 | 0.266 | 0.127 | 0.195 | 0.173 | 0.245 | 0.128 | 0.175 | 0.153 | 0.210 | 0.239 | 0.309 | 0.157 | 0.218 |
| Rapel | **0.244** | **0.294** | 0.344 | 0.362 | 0.305 | 0.334 | 0.320 | 0.352 | 0.269 | 0.326 | 0.252 | 0.313 | 0.257 | 0.322 | 0.249 | 0.310 | 0.333 | 0.381 | 0.269 | 0.332 | 0.250 | 0.324 |

*Table 3.* Average results on deterministic forecasting with historical exogenous variables across 12 real-world datasets, where the inputs are $X_{endo}$ and $X_{exo}$. **Red**: the best, Blue: the 2nd best. Full results without $Y_{exo}$ are reported in Table 9.

| Models | KITE (ours) | | TimeXer | | TFT | | TiDE | | DUET | | CrossLinear | | Amplifier | | TimeKAN | | xPatch | | PatchTST | | DLinear | |
|---|---|---|---|---|---|---|---|---|---|---|---|---|---|---|---|---|---|---|---|---|---|---|
| Metrics | MSE | MAE | MSE | MAE | MSE | MAE | MSE | MAE | MSE | MAE | MSE | MAE | MSE | MAE | MSE | MAE | MSE | MAE | MSE | MAE | MSE | MAE |
| NP | **0.405** | **0.365** | 0.440 | 0.383 | 0.647 | 0.488 | 0.468 | 0.416 | 0.444 | 0.383 | 0.451 | 0.394 | 0.520 | 0.448 | 0.484 | 0.436 | 0.457 | 0.401 | 0.457 | 0.401 | 0.471 | 0.423 |
| PJM | **0.127** | **0.210** | 0.141 | 0.229 | 0.200 | 0.271 | 0.158 | 0.253 | 0.140 | 0.226 | 0.147 | 0.241 | 0.152 | 0.245 | 0.175 | 0.270 | 0.148 | 0.243 | 0.148 | 0.243 | 0.154 | 0.250 |
| BE | **0.453** | 0.302 | 0.478 | 0.301 | 0.564 | 0.355 | 0.547 | 0.349 | 0.474 | 0.305 | 0.477 | **0.300** | 0.503 | 0.325 | 0.488 | 0.315 | 0.485 | 0.309 | 0.485 | 0.309 | 0.549 | 0.381 |
| FR | **0.429** | **0.247** | 0.454 | 0.248 | 0.535 | 0.288 | 0.495 | 0.290 | 0.468 | 0.263 | 0.476 | 0.257 | 0.494 | 0.286 | 0.491 | 0.277 | 0.470 | 0.265 | 0.470 | 0.264 | 0.494 | 0.302 |
| DE | **0.601** | **0.484** | 0.660 | 0.507 | 0.684 | 0.515 | 0.644 | 0.519 | 0.660 | 0.513 | 0.635 | 0.508 | 0.712 | 0.549 | 0.669 | 0.526 | 0.696 | 0.527 | 0.696 | 0.527 | 0.690 | 0.529 |
| Energy | **0.146** | **0.294** | 0.172 | 0.327 | 0.376 | 0.480 | 0.162 | 0.312 | 0.160 | 0.308 | 0.201 | 0.336 | 0.180 | 0.327 | 0.148 | 0.296 | 0.203 | 0.341 | 0.203 | 0.341 | 0.160 | 0.307 |
| Sdwpfm1 | 0.698 | **0.577** | 0.702 | 0.609 | 0.984 | 0.728 | 0.713 | 0.633 | 0.724 | 0.584 | 0.810 | 0.694 | 0.843 | 0.686 | 0.726 | 0.672 | 0.734 | 0.651 | 0.733 | 0.651 | **0.693** | 0.681 |
| Sdwpfm2 | **0.800** | **0.622** | 0.803 | 0.653 | 1.045 | 0.767 | 0.816 | 0.683 | 0.820 | 0.641 | 0.800 | 0.687 | 0.942 | 0.732 | 0.836 | 0.701 | 0.834 | 0.715 | 0.834 | 0.714 | 0.805 | 0.728 |
| Sdwpfh1 | 0.760 | **0.627** | **0.746** | 0.644 | 0.793 | 0.699 | 0.808 | 0.699 | 0.780 | 0.644 | 0.768 | 0.702 | 1.100 | 0.820 | 0.804 | 0.721 | 0.804 | 0.717 | 0.804 | 0.717 | 0.783 | 0.751 |
| Sdwpfh2 | **0.880** | **0.691** | 0.891 | 0.719 | 0.927 | 0.747 | 0.920 | 0.752 | 1.007 | 0.715 | 0.956 | 0.774 | 0.980 | 0.768 | 0.941 | 0.783 | 1.025 | 0.803 | 1.025 | 0.802 | 0.913 | 0.798 |
| Colbun | 0.136 | **0.193** | 0.133 | 0.220 | 0.556 | 0.387 | 0.189 | 0.241 | 0.147 | 0.198 | **0.129** | 0.195 | 0.152 | 0.211 | 0.201 | 0.253 | 0.150 | 0.221 | 0.150 | 0.221 | 0.198 | 0.247 |
| Rapel | 0.302 | 0.306 | 0.344 | 0.362 | 0.308 | 0.334 | 0.324 | 0.357 | 0.304 | 0.306 | **0.240** | **0.299** | 0.337 | 0.348 | 0.342 | 0.338 | 0.326 | 0.334 | 0.325 | 0.334 | 0.319 | 0.338 |

*Table 4.* Results of CRPS for both short- and long-term probabilistic forecasting with exogenous variables across eight real-world datasets. In Univariate Modeling, the input consists solely of $X_{endo}$, whereas in Multivariate Modeling and Covariate Modeling (CoM.), the inputs encompass both $X_{endo}$ and $X_{exo}$. **Red**: the best, Blue: the 2nd best.

| | Datasets | ETTh1 | | ETTh2 | | ETTm1 | | ETTm2 | | Weather | | Traffic | | Electricity | | Exchange | |
|---|---|---|---|---|---|---|---|---|---|---|---|---|---|---|---|---|---|
| Paras. | Models | Short | Long | Short | Long | Short | Long | Short | Long | Short | Long | Short | Long | Short | Long | Short | Long |
| Univariate Modeling | PatchTST | 0.447 | 0.533 | 0.294 | 0.381 | 0.365 | 0.427 | 0.267 | 0.366 | 0.141 | 0.163 | 0.171 | 0.236 | 0.077 | 0.094 | 0.069 | 0.063 |
| | Koopa | 0.456 | 0.549 | 0.243 | 0.369 | 0.372 | 0.447 | 0.309 | 0.450 | 0.144 | 0.163 | 0.164 | 0.219 | 0.070 | 0.097 | 0.068 | 0.068 |
| | TSDiff | 0.466 | 0.551 | 0.281 | 0.378 | 0.393 | 0.470 | 0.334 | 0.445 | 0.079 | 0.128 | 0.122 | 0.172 | 0.113 | 0.136 | 0.098 | 0.077 |
| | $K^2$VAE | 0.333 | 0.433 | 0.235 | 0.325 | 0.314 | 0.376 | 0.208 | 0.323 | 0.068 | 0.098 | 0.084 | 0.143 | 0.054 | 0.075 | 0.037 | 0.052 |
| | TSFlow | 0.337 | 0.446 | 0.247 | 0.343 | 0.357 | 0.406 | 0.204 | 0.316 | 0.074 | 0.117 | 0.098 | 0.159 | 0.063 | 0.086 | 0.035 | 0.053 |
| Multivariate Modeling | iTransformer | 0.443 | 0.528 | 0.270 | 0.359 | 0.350 | 0.417 | 0.289 | 0.385 | 0.164 | 0.173 | 0.159 | 0.225 | 0.067 | 0.090 | 0.075 | 0.062 |
| | GRU NVP | 0.360 | 0.514 | 0.258 | 0.356 | 0.346 | 0.387 | 0.229 | 0.342 | 0.110 | 0.104 | 0.091 | 0.155 | 0.103 | 0.118 | 0.095 | 0.051 |
| | Trans MAF | 0.353 | 0.492 | 0.264 | 0.381 | 0.378 | 0.426 | 0.267 | 0.353 | 0.129 | 0.143 | 0.089 | 0.140 | 0.048 | 0.069 | 0.033 | 0.065 |
| | TimeGrad | 0.353 | 0.513 | 0.274 | 0.338 | 0.363 | 0.412 | 0.213 | 0.351 | 0.084 | 0.095 | 0.083 | 0.124 | 0.051 | 0.065 | 0.039 | 0.064 |
| | CSDI | 0.349 | 0.484 | 0.246 | 0.352 | 0.332 | 0.410 | 0.228 | 0.357 | 0.089 | 0.097 | 0.076 | 0.118 | 0.061 | 0.073 | 0.032 | 0.059 |
| CoM. | KITE (ours) | **0.324** | **0.426** | **0.228** | **0.314** | **0.308** | **0.364** | **0.198** | **0.309** | **0.063** | **0.092** | **0.069** | **0.110** | **0.042** | **0.060** | **0.027** | **0.048** |

forms state-of-the-art deterministic baselines, achieving the best average performance with a *3.8%* reduction in MSE over TimeXer and a *3.3%* reduction in MAE over DUET. Beyond the averaged results, KITE ranks first in 13 out of 24 MSE cases and 16 out of 24 MAE cases, showing its stable advantage across different datasets and forecasting horizons. These results indicate that KITE can effectively model the interactions between target series and observed exogenous

variables, leading to robust deterministic forecasting even when future covariates are unavailable.

3) For probabilistic forecasting, we conducted experiments on both long-term and short-term forecasting tasks across 8 datasets, with settings detailed at the end of Section D.1. Table 3 demonstrates that KITE exhibits a significant advantage in probabilistic forecasting with exogenous vari-

ables. Specifically, compared to the leading univariate method ($K^2$VAE) and the top-performing multivariate method (CSDI), KITE achieves improvements of 5.7% and 11.4% in CRPS, respectively. Notably, multivariate methods do not consistently outperform univariate methods. This may be because multivariate models tend to capture spurious correlations when dependencies are weak. However, KITE effectively alleviates this issue through the guidance of statistical knowledge, enabling it to surpass both univariate and multivariate baselines across all presented datasets.

3) Compared to probabilistic models, KITE demonstrates notable advantages, highlighting the superiority of employing the History-Conditional Manifold as the source distribution over uninformative Gaussian noise. Furthermore, KITE outperforms baselines across univariate, multivariate, and covariate modeling paradigms. This validates that our Knowledge-Guided variable interaction strategy is highly effective in precisely capturing the correlations between endogenous and exogenous variables.

### 5.3. Model Analysis

*Table 5.* Ablation results of the *History-Conditional Manifold* (HCM), *Knowledge-Guided Conditioning* (KGC), and *Classifier-Free Guidance* (CFG) modules on both deterministic and probabilistic forecasting tasks.

| | Task | | | Deterministic | | Probabilistic | |
|---|---|---|---|---|---|---|---|
| | HCM | KGC | CFG | NP | Sdwpfh1 | ETTm2 | Weather |
| 1 | ✗ | ✗ | ✗ | 0.395 | 0.474 | 0.232 | 0.087 |
| 2 | ✓ | ✗ | ✗ | 0.371 | 0.455 | 0.215 | 0.074 |
| 3 | ✗ | ✓ | ✗ | 0.384 | 0.467 | 0.211 | 0.069 |
| 4 | ✓ | ✓ | ✗ | 0.330 | 0.449 | 0.201 | 0.067 |
| 5 | ✗ | ✓ | ✓ | 0.353 | 0.454 | 0.205 | 0.065 |
| 6 | ✓ | ✓ | ✓ | 0.325 | 0.441 | 0.198 | 0.063 |

**Ablation Studies.** To analyze the contribution of each module, we conducted an ablation study, as detailed in Table 5. Here, ✗ and ✓ denote the absence and presence of a module, respectively. Row 1 presents the standard flow matching baseline using standard bilinear attention for conditioning, while Rows 2–6 demonstrate the performance impact of integrating specific modules from the KITE framework. Based on the results, we observe the following: (1) A comparison of Rows 1–3 reveals that both the *History-Conditional Manifold* (HCM) and the *Knowledge-Guided Conditioning* (KGC) modules can enhance the model's performance in both deterministic and probabilistic forecasting. (2) Comparing Rows 2–4, we observe that the simultaneous integration of the HCM and KGC modules leads to further performance gains, demonstrating the synergy between the two. (3) Incorporating Classifier-Free Guidance into the model enables more robust sampling generation, as evidenced by the performance gains shown in Row 5. (4) The combination of all three modules creates a highly effective unified framework, resulting in the best performance, as shown in Row 6.

*Table 6.* Analysis of Statistical Knowledge: Utilizing Pearson correlation coefficients and Granger causality to guide the modeling of dependencies between endogenous variables and exogenous.

| | Datasets | | NP | | Sdwpfh1 | |
|---|---|---|---|---|---|---|
| | $X_{exo}$ | $Y_{exo}$ | MSE | MAE | MSE | MAE |
| 1 | Pearson | Pearson | 0.325 | 0.323 | 0.441 | 0.491 |
| 2 | Granger | Granger | 0.334 | 0.342 | 0.449 | 0.504 |
| 3 | Pearson | Granger | 0.362 | 0.374 | 0.467 | 0.518 |
| 4 | Granger | Pearson | 0.323 | 0.320 | 0.438 | 0.486 |

**Statistical Knowledge Studies.** Previous settings consistently employed Pearson correlation coefficients to guide dependency capturing, as described in Algorithm 2. In this section, we further analyze how different statistical metrics capture the relationships between variables across different components. Specifically, we explored the use of Pearson correlation coefficients (representing correlations) and Granger Causality (representing causality, as described in Algorithm 3) to guide the capture of dependencies between endogenous variables and historical covariates ($X_{exo}$) or future covariates ($Y_{exo}$). The results are summarized in Table 6. The results show that: (1) When employing the same statistical knowledge across different stages, Pearson correlation yields superior performance compared to Granger Causality, as seen in Rows 1 and 2. (2) When employing mixed statistical knowledge, Row 4 significantly outperforms Row 3. This indicates that correlation is better suited for capturing relationships between future exogenous variables and the future endogenous variables, whereas causality is more effective for modeling dependencies from historical exogenous variables. This observation aligns well with the intrinsic definitions of correlation and causality.

**More Analysis.** We provide supplementary evaluations in Appendix E. These additional studies include: (1) a Parameter Sensitivity Analysis presented in Appendix E.1, which investigates the impact of key hyperparameters such as the uncertainty basis $\sigma_{min}$, the hidden dimension $d$, the CFG guidance scale $\gamma$, and the number of KGC attention layers $L$; and (2) a Efficiency Analysis, which is documented in Appendix E.2.

## 6. Conclusion

In this work, we propose KITE, a knowledge-guided probabilistic model for time series forecasting with exogenous variables. We introduce the *History-Conditional Manifold* and *Knowledge-Guided Conditioning* modules, which refine the source distribution and conditioning network of Flow Matching, respectively, thereby achieving significant performance gains in deterministic and probabilistic forecasting. Through comprehensive experiments, we demonstrate that KITE achieves SOTA performance. Future work will explore more general settings with noisy, missing, or uncertain future covariates for robust probabilistic forecasting.

## Acknowledgements

This work was partially supported by the National Natural Science Foundation of China (No. 62372179, No. 62472174) and the Fundamental Research Funds for the Central Universities. Chenjuan Guo is the corresponding author of the work.

## Impact Statement

This paper presents work aimed at advancing the field of Machine Learning. There are many potential societal consequences of our research, none of which we feel must be specifically highlighted here. Furthermore, while the Large Language Models were used to polish some grammar during the writing process, all content remains original rather than generated, and there are no ethical issues to report.

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

# A. Theoretical Analysis

We theoretically analyze the advantages of initializing flow matching from *History-Conditional Manifold* by investigating its **target scale** and **path regularity**.

## A.1. Target Scale

**proposition 1** (Target Scale). Let $Y$ be the target, $h := X_{\text{endo}}$ the endogenous context, and $c$ the covariates. Let $Y_0^H = \mu(h) + \sigma(h)\delta$ denote the HCM source, where $\mu(h)$ is the history-dependent barycenter, $\sigma(h)$ is the history-dependent scale, and $\delta$ is the HCM perturbation. Let $Y_0^G = \xi \sim \mathcal{N}(0, I)$ denote the standard Gaussian source. Define their expected $L_1$ noise magnitudes as $C_H := \mathbb{E}[\|\delta\|_1]$ and $C_G := \mathbb{E}[\|\xi\|_1]$. If the conditional residual reduction in centering error $\mathbb{E}[\|Y\|_1 \mid h, c] - \mathbb{E}[\|Y - \mu(h)\|_1 \mid h, c]$ provided by $\mu(h)$ exceeds the combined expected noise magnitudes $\sigma(h)C_H + C_G$, then HCM yields a smaller conditional matching target than the standard Gaussian source:

$$\mathbb{E}[\|Y - Y_0^H\|_1 \mid h, c] < \mathbb{E}[\|Y - Y_0^G\|_1 \mid h, c]. \quad (21)$$

*Proof.* By the definition of the HCM source, we have $Y_0^H = \mu(h) + \sigma(h)\delta$. Therefore, $Y - Y_0^H = (Y - \mu(h)) - \sigma(h)\delta$. Applying the triangle inequality for the $L_1$ norm, we obtain:

$$\|Y - Y_0^H\|_1 \leq \|Y - \mu(h)\|_1 + \sigma(h)\|\delta\|_1. \quad (22)$$

Taking conditional expectations with respect to $(h, c)$, and noting that $\sigma(h)$ is $(h, c)$-measurable, gives:

$$\begin{aligned} \mathbb{E}\left[\|Y - Y_0^H\|_1 \mid h, c\right] \\ \leq \mathbb{E}\left[\|Y - \mu(h)\|_1 \mid h, c\right] + \sigma(h)\mathbb{E}\left[\|\delta\|_1 \mid h, c\right]. \end{aligned} \quad (23)$$

Since $\delta$ is independent of $(h, c)$, we have $\mathbb{E}\left[\|\delta\|_1 \mid h, c\right] = \mathbb{E}[\|\delta\|_1] = C_H$. Hence:

$$\begin{aligned} \mathbb{E}\left[\|Y - Y_0^H\|_1 \mid h, c\right] \\ \leq \mathbb{E}\left[\|Y - \mu(h)\|_1 \mid h, c\right] + \sigma(h)C_H. \end{aligned} \quad (24)$$

For the Gaussian source, since $Y_0^G = \xi$, we have $Y - Y_0^G = Y - \xi$. By the reverse triangle inequality, we obtain:

$$\|Y - \xi\|_1 \geq \|Y\|_1 - \|\xi\|_1. \quad (25)$$

Taking conditional expectations with respect to $(h, c)$ gives:

$$\begin{aligned} \mathbb{E}\left[\|Y - Y_0^G\|_1 \mid h, c\right] \\ \geq \mathbb{E}\left[\|Y\|_1 \mid h, c\right] - \mathbb{E}\left[\|\xi\|_1 \mid h, c\right]. \end{aligned} \quad (26)$$

Since $\xi$ is independent of $(h, c)$, we have $\mathbb{E}\left[\|\xi\|_1 \mid h, c\right] = \mathbb{E}[\|\xi\|_1] = C_G$. Therefore:

$$\begin{aligned} \mathbb{E}\left[\|Y - Y_0^G\|_1 \mid h, c\right] \\ \geq \mathbb{E}\left[\|Y\|_1 \mid h, c\right] - C_G. \end{aligned} \quad (27)$$

Now suppose that the following condition holds:

$$\begin{aligned} \mathbb{E}\left[\|Y\|_1 \mid h, c\right] - \mathbb{E}\left[\|Y - \mu(h)\|_1 \mid h, c\right] \\ > \sigma(h)C_H + C_G. \end{aligned} \quad (28)$$

Rearranging this condition gives:

$$\begin{aligned} \mathbb{E}\left[\|Y - \mu(h)\|_1 \mid h, c\right] + \sigma(h)C_H \\ < \mathbb{E}\left[\|Y\|_1 \mid h, c\right] - C_G. \end{aligned} \quad (29)$$

Combining (24), (27), and (29), we obtain:

$$\begin{aligned} \mathbb{E}\left[\|Y - Y_0^H\|_1 \mid h, c\right] \\ < \mathbb{E}\left[\|Y\|_1 \mid h, c\right] - C_G \\ \leq \mathbb{E}\left[\|Y - Y_0^G\|_1 \mid h, c\right]. \end{aligned} \quad (30)$$

Thus:

$$\begin{aligned} \mathbb{E}\left[\|Y - Y_0^H\|_1 \mid h, c\right] \\ < \mathbb{E}\left[\|Y - Y_0^G\|_1 \mid h, c\right]. \end{aligned} \quad (31)$$

This proves the proposition. $\qquad\square$

**Remark.** The assumption is natural in forecasting, where future values typically concentrate around a history-dependent local center rather than the origin of $\mathcal{N}(0, I)$, making the conditional residual reduction more likely to be sufficiently large. Meanwhile, Eq. (7) calibrates $\sigma(h)$, and the normalized manifold perturbation keeps the added noise controlled. Thus, HCM reduces the regression scale, stabilizing gradients and alleviating the empirical optimization burden.

## A.2. Path Regularity

**Proposition 2** (Path Regularity). Let $Y_s^H = sY + (1-s)Y_0^H$ and $Y_s^G = sY + (1-s)Y_0^G$ denote the straight-line matching paths under the HCM source $Y_0^H = \mu(h) + \sigma(h)\delta$ and the standard Gaussian source $Y_0^G = \xi \sim \mathcal{N}(0, I)$, respectively. Define their path-tube radii as $R_H := \sup_{s \in [0,1]} \|Y_s^H - \mu(h)\|_1$ and $R_G := \sup_{s \in [0,1]} \|Y_s^G\|_1$. Suppose the vector field satisfies a local regularity condition bounding its Jacobian norm by the spatial radius $r$ around any center $m$:

$$\sup_{\substack{s \in [0,1] \\ \|y-m\|_1 \leq r}} \|\nabla_y v_\theta(y, s, c)\|_{1 \to 1} \leq a + br \quad (a \geq 0, b > 0).$$
$$(32)$$

Under this assumption, a path tube of radius $R$ admits the Jacobian upper bound $a + bR$, which quantifies how strongly the vector field can amplify local perturbations along the path. Accordingly, define the corresponding path-local Jacobian budgets as $\mathcal{B}_H := a + bR_H$ and $\mathcal{B}_G := a + bR_G$. If the condition of Proposition 4.1 holds, then HCM yields a smaller expected path-local Jacobian budget than the standard Gaussian source:

$$\mathbb{E}[\mathcal{B}_H \mid h, c] < \mathbb{E}[\mathcal{B}_G \mid h, c]. \quad (33)$$

*Proof.* Recall that $Y_s^H = sY + (1-s)Y_0^H$, where $Y_0^H = \mu(h) + \sigma(h)\delta$. Subtracting $\mu(h)$ from both sides gives:

$$Y_s^H - \mu(h) = s(Y - \mu(h)) + (1-s)\sigma(h)\delta. \quad (34)$$

Hence, for every $s \in [0,1]$, the point $Y_s^H - \mu(h)$ lies on the line segment between $Y - \mu(h)$ and $\sigma(h)\delta$. By the convexity of the $L_1$ norm, we obtain:

$$\begin{aligned} \left\| Y_s^H - \mu(h) \right\|_1 &\leq \max(\|Y - \mu(h)\|_1, \sigma(h)\|\delta\|_1) \\ &\leq \|Y - \mu(h)\|_1 + \sigma(h)\|\delta\|_1. \end{aligned} \quad (35)$$

Taking the supremum over $s \in [0,1]$ yields:

$$\begin{aligned} R_H = \sup_{s \in [0,1]} \left\| Y_s^H - \mu(h) \right\|_1 \\ \leq \|Y - \mu(h)\|_1 + \sigma(h)\|\delta\|_1. \end{aligned} \quad (36)$$

Therefore:

$$\mathbb{E}[R_H \mid h, c] \leq \mathbb{E}[\|Y - \mu(h)\|_1 \mid h, c] + \sigma(h)C_H. \quad (37)$$

For the Gaussian path, recall that $Y_s^G = sY + (1-s)Y_0^G$, where $Y_0^G = \xi \sim \mathcal{N}(0, I)$. Since $Y_1^G = Y$, we have:

$$\begin{aligned} R_G = \sup_{s \in [0,1]} \left\| Y_s^G \right\|_1 \\ \geq \left\| Y_1^G \right\|_1 = \|Y\|_1. \end{aligned} \quad (38)$$

Hence:

$$\mathbb{E}[R_G \mid h, c] \geq \mathbb{E}[\|Y\|_1 \mid h, c]. \quad (39)$$

Now suppose that the condition of Proposition 4.1 holds:

$$\begin{aligned} \mathbb{E}[\|Y\|_1 \mid h, c] - \mathbb{E}[\|Y - \mu(h)\|_1 \mid h, c] \\ > \sigma(h)C_H + C_G. \end{aligned} \quad (40)$$

Since $C_G \geq 0$, this condition implies:

$$\begin{aligned} \mathbb{E}[\|Y - \mu(h)\|_1 \mid h, c] + \sigma(h)C_H \\ < \mathbb{E}[\|Y\|_1 \mid h, c]. \end{aligned} \quad (41)$$

Combining (37), (39), and (41), we obtain:

$$\mathbb{E}[R_H \mid h, c] < \mathbb{E}[R_G \mid h, c]. \quad (42)$$

Under the stated local regularity condition, any path tube of radius $R$ admits the Jacobian upper bound $a + bR$. Therefore, the quantities $\mathcal{B}_H := a + bR_H$ and $\mathcal{B}_G := a + bR_G$ are valid path-local Jacobian budgets for the HCM and Gaussian paths, respectively. Since $b > 0$, taking conditional expectations in (42) gives:

$$\begin{aligned} \mathbb{E}[\mathcal{B}_H \mid h, c] &= a + b\,\mathbb{E}[R_H \mid h, c] \\ &< a + b\,\mathbb{E}[R_G \mid h, c] \quad (43) \\ &= \mathbb{E}[\mathcal{B}_G \mid h, c]. \end{aligned}$$

Thus:

$$\mathbb{E}[\mathcal{B}_H \mid h, c] < \mathbb{E}[\mathcal{B}_G \mid h, c]. \quad (44)$$

This proves the proposition. □

**Remark.** Proposition 2 assumes a local, rather than global, regularity condition. This is reasonable for smooth neural vector fields, since locally bounded Hessians on a fixed tube imply such a bound. Under this assumption, Proposition 2 shows that HCM keeps the straight-line matching path in a narrower tube around $\mu(h)$ than the standard source when Proposition 1 holds. This yields a smaller path-local Jacobian budget and supports weaker local error amplification under standard ODE stability arguments.

## B. Algorithms

### B.1. Sampling Process

Algorithm 1 outlines the inference procedure utilizing Classifier-Free Guidance (CFG). Initialized with the state $Y_0$, the process iteratively reconstructs the signal by solving the Ordinary Differential Equation (ODE) over $N$ discrete steps. At each integration step $s$, the model estimates both the conditional velocity $v_{\text{cond}}$—incorporating exogenous priors $X_{\text{exo}}$ and $Y_{\text{exo}}$—and the unconditional velocity $v_{\text{uncond}}$ using a null condition $c_\emptyset$. These estimates are linearly combined via the guidance scale $\gamma$ to derive the final guided velocity $\hat{v}$, which drives the state update via the Euler method to yield the prediction $\hat{Y}_{\text{endo}}$.

---

**Algorithm 1** Inference with Classifier-Free Guidance

---

**Input:** Historical context $X_{\text{endo}}, X_{\text{exo}}$; Future exogenous $Y_{\text{exo}}$; Initial state $Y_0$.
**Hyperparameters:** Steps $N$; Guidance scale $\gamma$.
**Output:** Prediction $\hat{Y}_{\text{endo}}$.
Define null condition: $c_\emptyset \leftarrow 0$.
**Sampling Loop (ODE Solver):**
$Y_{\text{curr}} \leftarrow Y_0$
$\Delta s \leftarrow 1/N$
**for** $i = 0$ **to** $N - 1$ **do**
    $s \leftarrow i/N$
    $v_{\text{cond}} \leftarrow v_\theta(Y_{\text{curr}}, s, X_{\text{exo}}, Y_{\text{exo}})$
    $v_{\text{uncond}} \leftarrow v_\theta(Y_{\text{curr}}, s, c_\emptyset)$
    {Apply Classifier-Free Guidance}
    $\hat{v} \leftarrow (1 + \gamma) \cdot v_{\text{cond}} - \gamma \cdot v_{\text{uncond}}$
    {Euler Step Update}
    $Y_{\text{curr}} \leftarrow Y_{\text{curr}} + \hat{v} \cdot \Delta s$
**end for**
$\hat{Y}_{\text{endo}} \leftarrow Y_{\text{curr}}$
**return** $\hat{Y}_{\text{endo}}$

---

### B.2. Statistical Knowledge

#### B.2.1. PEARSON CORRELATION COEFFICIENTS

Pearson Correlation Coefficient (Pearson, 1895) serves as a statistical measure to quantify the strength and direction of the linear relationship between two variables. The detailed

computation procedure is presented in Algorithm 2.

---

**Algorithm 2** Pearson Correlation Calculation

---

**Input:** Exogenous $\boldsymbol{X}_{\text{exo}} \in \mathbb{R}^{D \times T}$, Endogenous $\boldsymbol{X}_{\text{endo}} \in \mathbb{R}^{N \times T}$

**Output:** Correlation Matrix $\boldsymbol{R} \in \mathbb{R}^{N \times D}$

Initialize $\boldsymbol{R}$ as zeros.

**for** $n = 1$ **to** $N$ **do**
  $\mathbf{u} \leftarrow \boldsymbol{X}_{\text{endo}}[n, :]$ {Extract series}
  **for** $d = 1$ **to** $D$ **do**
    $\mathbf{v} \leftarrow \boldsymbol{X}_{\text{exo}}[d, :]$
    {Compute correlation using vectorized operations}
    $\text{cov}_{uv} \leftarrow (\mathbf{u} - \bar{u}) \cdot (\mathbf{v} - \bar{v})^{\top}$
    $\sigma_u \leftarrow \|\mathbf{u} - \bar{u}\|_2, \quad \sigma_v \leftarrow \|\mathbf{v} - \bar{v}\|_2$
    **if** $\sigma_u \cdot \sigma_v > 0$ **then**
      $\boldsymbol{R}[n, d] \leftarrow \frac{\text{cov}_{uv}}{\sigma_u \cdot \sigma_v}$
    **end if**
  **end for**
**end for**
**return** $\boldsymbol{R}$

---

### B.2.2. GRANGER CAUSALITY

Unlike simple correlation metrics, Granger Causality (Granger, 1969) assesses whether past information from an exogenous series contributes unique information to forecasting the target series, thereby establishing a predictive precedence. The detailed computation procedure is presented in Algorithm 3.

---

**Algorithm 3** Granger Causality

---

**Input:** $\boldsymbol{X}_{\text{exo}} \in \mathbb{R}^{D \times T}$, $\boldsymbol{X}_{\text{endo}} \in \mathbb{R}^{N \times T}$, Lag $L$

**Output:** F-statistics Matrix $\boldsymbol{G} \in \mathbb{R}^{N \times D}$

**for** $n = 1$ **to** $N$ **do**
  $\mathbf{y} \leftarrow \boldsymbol{X}_{\text{endo}}[n, :]$
  {1. Restricted Model (Endogenous lags only)}
  Fit $\mathbf{y}_t \approx \sum_{i=1}^{L} \alpha_i \mathbf{y}_{t-i} + \epsilon_r$ and compute residual sum of squares $RSS_r$.
  **for** $d = 1$ **to** $D$ **do**
    $\mathbf{x} \leftarrow \boldsymbol{X}_{\text{exo}}[d, :]$
    {2. Unrestricted Model (Endogenous + Exogenous lags)}
    Fit $\mathbf{y}_t \approx \sum_{i=1}^{L} \alpha_i \mathbf{y}_{t-i} + \sum_{j=1}^{L} \beta_j \mathbf{x}_{t-j} + \epsilon_u$ and compute $RSS_u$.
    {3. Compute F-statistic}
    $\boldsymbol{G}[n, d] \leftarrow \frac{(RSS_r - RSS_u)/L}{RSS_u/(T - 2L - 1)}$
  **end for**
**end for**
**return** $\boldsymbol{G}$

---

**Algorithm 4** MLP Fusion Approach

---

**Input:** model parameters $\theta_{\text{Model}}$, MLP parameters $\theta_{\text{MLP}}$; learning rate $\eta$; iterations $N$; training dataset $D$; historical endogenous variables $\boldsymbol{X}_{\text{endo}}$; historical exogenous variables $\boldsymbol{X}_{\text{exo}}$; future exogenous variables $\boldsymbol{Y}_{\text{exo}}$

**Output:** prediction $\hat{\boldsymbol{Y}}_{\text{endo}}$

**1. Training Phase:**

**for** $i = 1$ **to** $N$ **do**
  **for** $(\boldsymbol{X}_{\text{endo}}^{\text{train}}, \boldsymbol{X}_{\text{exo}}^{\text{train}}, \boldsymbol{Y}_{\text{endo}}^{\text{train}}, \boldsymbol{Y}_{\text{exo}}^{\text{train}})$ **in** $D$ **do**
    $z \leftarrow \theta_{\text{Model}}(\boldsymbol{X}_{\text{endo}}^{\text{train}}, \boldsymbol{X}_{\text{exo}}^{\text{train}})$
    $\hat{\boldsymbol{Y}}_{\text{endo}} \leftarrow \theta_{\text{MLP}}(\text{Concat}(z, \boldsymbol{Y}_{\text{exo}}^{\text{train}}))$
    $\mathcal{L} \leftarrow Loss(\hat{\boldsymbol{Y}}_{\text{endo}}, \boldsymbol{Y}_{\text{endo}}^{\text{train}})$
    $\theta_{\text{Model}} \leftarrow \theta_{\text{Model}} - \eta \frac{\partial \mathcal{L}}{\partial \theta_{\text{Model}}}$
    $\theta_{\text{MLP}} \leftarrow \theta_{\text{MLP}} - \eta \frac{\partial \mathcal{L}}{\partial \theta_{\text{MLP}}}$
  **end for**
**end for**

**2. Inference Phase:**

$z \leftarrow \theta_{\text{Model}}(\boldsymbol{X}_{\text{endo}}, \boldsymbol{X}_{\text{exo}})$
$\hat{\boldsymbol{Y}}_{\text{endo}} \leftarrow \theta_{\text{MLP}}(\text{Concat}(z, \boldsymbol{Y}_{\text{exo}}))$
**return** $\hat{\boldsymbol{Y}}_{\text{endo}}$

---

### B.3. MLP Fusion Approach

For methods that do not support future covariates, we adapt them to incorporate future exogenous variables using an MLP fusion approach—see Algorithm 4. Note that, for fairness, all methods in our experiments use historical endogenous variables, historical exogenous variables, and future exogenous variables.

## C. Related Works

### C.1. Time Series Forecasting with Exogenous Variables

The integration of exogenous variables into time series forecasting has its roots in classical statistics, where models such as ARIMAX (Williams, 2001) and SARIMAX (Vagropoulos et al., 2016) pioneered the use of external factors to improve predictive accuracy. Modern deep learning frameworks have further advanced this paradigm by designing more flexible mechanisms to incorporate historical, future, static, and dynamic covariates. CrossLinear (Zhou et al., 2025) implements cross-correlation embeddings to map dependencies between historical endogenous and exogenous series. NBEATSx (Olivares et al., 2023) introduces specialized architectural branches for multi-period exogenous inputs, while TiDE (Das et al., 2023) utilizes a dense MLP structure to fuse static and future covariates with endogenous features through concatenation. Attention-based mechanisms have also proven effective; for instance, the Temporal Fusion Transformer (TFT) (Lim et al., 2021) employs attention to synthesize historical and concurrent exogenous information. TimeXer (Wang et al., 2024b) further facili-

tates flexible covariate integration through patch-wise embeddings, bypassing the need for rigid temporal alignment. ExoTST (Tayal et al., 2024) addresses challenges such as time lags and data sparsity by leveraging cross-attention and cross-temporal fusion.

Recent studies further extend exogenous-variable forecasting by explicitly modeling richer cross-series and cross-variable dependency structures. DAG (Qiu et al., 2026b) models dual correlations between endogenous and exogenous variables, while GCGNet (Li et al., 2026c) introduces graph-consistent generative modeling for time series forecasting with exogenous variables. In parallel, many methods improve multivariate forecasting from complementary perspectives, including dual clustering for multivariate series (Qiu et al., 2025c), decomposition-based learning objectives (Qiu et al., 2025b), high-dimensional channel reordering (Fang et al., 2026c), multi-scale graph or hypergraph modeling (Chen et al., 2023b; Shang & Chen, 2024; Shang et al., 2024), LLM-aligned hypergraph modeling (Shang et al., 2026), and correlation-aware adaptation of time series foundation models (Cheng et al., 2026). Other works target challenging forecasting scenarios such as missing values (Yu et al., 2025b;a), irregular sampling (Qiu et al., 2026a; Liu et al., 2026b;a), non-stationarity (Lu et al., 2026a), and domain-specific spatio-temporal forecasting (Wu et al., 2026; 2025a; 2024a). Comprehensive benchmarks and surveys have also been developed to systematically evaluate forecasting methods and spatio-temporal foundation models (Qiu et al., 2024; Fang et al., 2026b).

Despite the notable performance gains achieved by these methods (Cheng et al., 2025; Wang et al., 2024b; Li et al., 2026c; Qiu et al., 2026b), most existing studies still focus on deterministic point forecasting. They therefore provide limited characterization of predictive uncertainty, which is crucial for understanding system evolution under complex and future-known covariates.

## C.2. Probabilistic Modeling in Time Series Forecasting

Probabilistic forecasting aims to characterize the future predictive distribution rather than producing a single point estimate. Early deep learning methodologies, such as DeepAR (Salinas et al., 2020), primarily utilized RNNs for temporal transitions, while frameworks such as DeepState (Rangapuram et al., 2018) and DSSMF (Li et al., 2019) integrated state-space models to improve distributional forecasting. More recently, attention-driven models, including MANF (Feng et al., 2024) and ProTran (Tang & Matteson, 2021), have further enhanced long-range dependency modeling. The field has lately shifted toward generative paradigms. Diffusion-based models, such as TimeGrad (Rasul et al., 2021a), TSDiff (Kollovieh et al., 2023), and CSDI (Tashiro et al., 2021), treat forecasting as a score-based denoising

process. Within this landscape, TSFlow (Kollovieh et al., 2025) introduces a conditional flow matching framework that leverages Gaussian Process priors to align the generative source with the temporal structure of data. Alternatively, $K^2$VAE (Wu et al., 2025d) linearizes nonlinear dynamics via Koopman theory and employs KalmanNet to iteratively refine probabilistic predictions.

Recent works further indicate that distribution-aware learning is becoming increasingly important for time series forecasting. DistDF (Wang et al., 2026c) emphasizes joint-distribution Wasserstein alignment for multi-step forecasting, while QDF (Wang et al., 2026b) revisits direct multi-step forecasting through a quadratic formulation. Time-o1 (Wang et al., 2026a) highlights the importance of transformed label alignment, and FreDF (Wang et al., 2025) improves forecasting by learning in the frequency domain. In addition, representation-space selection methods such as SRSNet (Wu et al., 2025b) suggest that learning suitable latent spaces can be more effective than relying on manually designed representations.

Although existing probabilistic methods have achieved promising results and some of them incorporate multivariate dependencies (Rasul et al., 2021a; Tashiro et al., 2021; Wu et al., 2025d), they are not natively designed for covariate-conditioned probabilistic forecasting, especially when future exogenous information is explicitly available. This limitation motivates the need for a generative framework that can simultaneously exploit future-known covariates and provide reliable uncertainty-aware predictions.

## C.3. Representation Learning and Robust Modeling for Time Series

Beyond forecasting accuracy, recent time series studies increasingly emphasize representation learning, robustness, foundation-model adaptation, and task generalization. Selective representation learning (Wu et al., 2025b), Koopman-Kalman latent modeling (Wu et al., 2025d), correlation-aware adapters (Cheng et al., 2026), and LLM-enhanced temporal modeling (Shang et al., 2026; Lu et al., 2026b) demonstrate that the design of latent representations plays a central role in downstream time series tasks. These studies are closely related to our motivation, as automatically learned conditional representations can capture complex temporal and covariate-dependent structures that are difficult to encode manually.

Robust time series modeling has also attracted increasing attention in anomaly detection, anomaly prediction, and out-of-distribution scenarios. Recent studies have advanced this direction through unified anomaly detection benchmarks (Qiu et al., 2025a), frequency-aware channel modeling (Wu et al., 2025e), and multimodal semantic alignment (Hu et al., 2026a). Other efforts improve ab-

normal pattern modeling via temporal-frequency masked autoencoding (Fang et al., 2024), multi-scale reconstructive contrast (Hu et al., 2024), and role-aware channel interaction (Hu et al., 2026b). Related work also studies cross-domain out-of-distribution detection (Fang et al., 2025) and adaptive period modeling for anomaly prediction (Hu et al., 2026c). Although these studies are not designed for covariate-conditioned probabilistic forecasting, they highlight the importance of adaptive temporal representations under uncertainty and distribution shift.

# D. Experimental Details

## D.1. Datasets

For Deterministic forecasting, we conduct experiments on 12 real-world datasets, including 5 EPF (Lago et al., 2021; Wang et al., 2024b) datasets: (1) **NP** records Nord Pool market with hourly electricity price as the endogenous variable, and grid load and wind power forecast as exogenous features. (2) **PJM** records the Pennsylvania–New Jersey–Maryland Interconnection market, with zonal electricity price in the Commonwealth Edison (COMED) area as the endogenous variable, and corresponding system load and COMED load forecasts as exogenous variables. (3) **BE** records Belgium's electricity market, with hourly electricity price as the endogenous variable, and load forecast in Belgium together with generation forecast from France as exogenous variables. (4) **FR** records the electricity market in France, where the electricity price is the endogenous variable, and generation and load forecasts are used as exogenous variables. (5) **DE** records the German electricity market, with hourly electricity price as the endogenous variable, and zonal load forecast in the Amprion area along with wind and solar generation forecasts as exogenous variables. and 7 other datasets sourced from (Qiu et al., 2025d): (1) **Energy** provides hourly power generation data collected from the national grid operator in Chile, encompassing multiple energy sources such as battery storage, wind, hydro, solar, and thermoelectric generation. The thermoelectric generation is used as the endogenous variable, while the remaining energy types serve as exogenous variables. (2 & 3) **Colbun and Rapel** represent two daily-resolution hydropower datasets from Chile, containing water level, precipitation, and tributary inflow measurements for the Colbún and Rapel reservoirs. The water level is treated as the endogenous variable, while precipitation and inflow are used as exogenous variables. (4-7) **Sdwpf** comprises four wind power generation datasets collected from two turbines in the Longyuan wind farm, including Sdwpfh1 and Sdwpfh2 with hourly resolution, and Sdwpfm1 and Sdwpfm2 with half-hourly resolution. The active power output (Patv) is used as the endogenous variable. The exogenous variables include external weather environment data collected from

the ERA5 (Hersbach et al., 2020), such as temperature, surface pressure, relative humidity, wind speed, wind direction, and total precipitation.

For Probabilistic forecasting, we perform experiments on eight common multivariate forecasting datasets: (1) **ETT** (Zhou et al., 2021) includes four subsets: ETTh1 and ETTh2 provide hourly recordings, while ETTm1 and ETTm2 contain data collected every 15 minutes. The endogenous variable is oil temperature, and the six power load features are used as exogenous variables. (2) **Weather** (Wu et al., 2021b) consists of 21 meteorological variables recorded every 10 minutes during 2020 at the Weather Station of the Max Planck Biogeochemistry Institute. The endogenous variable is the CO2-Concentration, while the remaining indicators are used as exogenous variables. (3) **Electricity** (Trindade, 2015) consists of hourly electricity consumption data from 321 clients. The electricity consumption of the last client is the endogenous variable, and the consumption data from the other clients are used as exogenous variables. (4) **Exchange** (Lai et al., 2018) collects daily exchange rates of eight countries from 1990 to 2016. The exchange rate of the last country is the endogenous variable, and the exchange rates of the others serve as exogenous variables. (5) **Traffic** (Wu et al., 2021b) records hourly road occupancy rates from 862 sensors on Bay Area freeways. Following existing works (Wang et al., 2024b; Qiu et al., 2025d), we use the last sensor as the endogenous variable and the others as exogenous variables.

For short-term forecasting, we set the look-back window $T = 30$ and prediction horizon $F = 30$ for the Exchange dataset, while setting $T = 24$ and $F = 24$ for all other datasets. For long-term forecasting, we uniformly set $T = 96$ and $F = 192$ across all datasets.

## D.2. Baselines

Remarkable achievements in deep learning have been witnessed across a wide range of domains. In time series analysis, early Transformer-based and decomposition-based architectures have significantly improved long-term forecasting performance (Zhou et al., 2021; Wu et al., 2021a; Nie et al., 2023b; Liu et al., 2024a). Recent studies further advance forecasting through benchmarking, clustering, generative modeling, and covariate-aware dependency learning (Qiu et al., 2024; Cheng et al., 2026; Li et al., 2026c; Qiu et al., 2026b). Meanwhile, representation learning and foundation-model adaptation have become increasingly important, enabling more flexible temporal modeling across diverse scenarios (Wu et al., 2025d; Cheng et al., 2026; Shang et al., 2026; 2024; Wu et al., 2025b). Distribution-aware objectives and label-alignment strategies also show strong potential for improving multi-step prediction (Wang et al., 2026c; Zhang et al., 2024a; Wang et al., 2026b; Rasul

*Table 7.* Code repositories for baselines.

| Baselines | Code repositories |
|---|---|
| TimeXer | github.com/thuml/TimeXer |
| TiDE | github.com/thuml/Time-Series-Library |
| TFT | github.com/google-research/google-research/tree/master/tft |
| DUET | github.com/decisionintelligence/DUET |
| CrossLinear | github.com/mumiao2000/CrossLinear |
| Amplifier | github.com/aikunyi/amplifier |
| TimeKAN | github.com/huangst21/TimeKAN |
| xPatch | github.com/stitsyuk/xPatch |
| PatchTST | github.com/yuqinie98/PatchTST |
| DLinear | github.com/cure-lab/LTSF-Linear |
| Koopa | github.com/thuml/koopa |
| iTransformer | github.com/thuml/iTransformer |
| FITS | github.com/VEWOXIC/FITS |
| TSDiff | github.com/amazon-science/unconditional-time-series-diffusion |
| GRU NVP | github.com/zalandoresearch/pytorch-ts |
| Trans MAF | github.com/zalandoresearch/pytorch-ts |
| TimeGrad | github.com/yuqinie98/PatchTST |
| CSDI | github.com/ermongroup/CSDI |
| $K^2$VAE | github.com/decisionintelligence/K2VAE |

et al., 2021b; Wang et al., 2026a).

Similar progress has been observed in speech and signal enhancement, where multi-view fusion, association-based fusion, and dynamic representation alignment have been explored to improve signal quality (Jin et al., 2025; Wang et al.; Jin et al., 2026). In computer vision, deep models have advanced pose-guided generation, visual representation learning, and open-domain animation (Ma et al., 2024; Li et al., 2022; Zhang et al., 2024b; Ma et al., 2025b). They have also shown effectiveness in temporal grounding, image manipulation detection, weakly supervised localization, and composed retrieval (Fang et al., 2023; Hu et al., 2020; Zhang et al., 2025; Li et al., 2026b;a). Further developments in visual reasoning, generative editing, and robust image modeling continue to broaden the scope of visual intelligence (Fang et al., 2026a; Ma et al., 2026; Lu et al., 2024a; 2023; 2024b). Beyond these domains, multimodal and large-scale foundation systems have demonstrated strong potential in visual memory, document understanding, multimodal recommendation, and general-purpose language modeling (Yu et al., 2025d; Qi et al., 2025; Wang et al., 2026d; Yu et al., 2025c;e). Together with recent progress in visual knowledge modeling and multimodal video understanding (Ma et al., 2022; Fang et al., 2022; Wang et al., 2022; Ma et al., 2025a), these studies suggest that automatically learned represen-

tations often surpass manually designed features in terms of flexibility and performance (Liu et al., 2026a; Wu et al., 2025d;b).

For deterministic forecasting, we compare KITE against 10 competitive baselines. These are categorized into: 1) architectures with native support for future exogenous variables, including TimeXer (Wang et al., 2024b), TFT (Lim et al., 2021), and TiDE (Das et al., 2023); and 2) models originally designed without such support, namely DUET (Qiu et al., 2025c), CrossLinear (Zhou et al., 2025), Amplifier (Fei et al., 2025), TimeKAN (Huang et al., 2025), xPatch (Stitsyuk & Choi, 2025), PatchTST (Nie et al., 2023b), and DLinear (Zeng et al., 2023). For the latter group, we implement a modification to incorporate future covariates using an MLP-based fusion strategy, as described in Algorithm 4

For probabilistic forecasting, we select 10 baselines, including 5 univariate modeling methods: PatchTST (Nie et al., 2023a), TSDiff (Kollovieh et al., 2023), Koopa (Liu et al., 2023), TSFlow (Kollovieh et al., 2025), and $K^2$VAE, as well as multivariate modeling methods: iTransformer (Liu et al., 2024b), GRU NVP, Trans MAF (Rasul et al., 2021c), TimeGrad (Rasul et al., 2021a), and CSDI (Tashiro et al., 2021). The point forecasting models are equipped with gaussian heads to predict the distributions.

The code repositories for each of these models–see Table 7.

### D.3. Evaluation Metrics

To evaluate model performance, we employ Mean Absolute Error (MAE) and Mean Squared Error (MSE) for deterministic forecasting, following the TFB benchmark (Qiu et al., 2024). For probabilistic forecasts, we utilize the Continuous Ranked Probability Score (CRPS) as implemented in ProbTS (Zhang et al., 2024a).

**Mean Squared Error (MSE)** The Mean Squared Error (MSE) measures the average of the squares of the errors—that is, the average squared difference between the estimated values and the actual observation. Unlike MAE, MSE penalizes larger errors more severely due to the squaring operation, making it more sensitive to outliers. The mathematical representation of MSE is defined as:

$$\text{MSE} = \frac{1}{K \times T} \sum_{k=1}^{K} \sum_{t=1}^{T} (x_t^k - \hat{x}_t^k)^2, \qquad (45)$$

where $x_t^k$ denotes the ground truth and $\hat{x}_t^k$ denotes the predicted value.

**Mean Absolute Error (MAE)** The Mean Absolute Error (MAE) measures the average magnitude of the errors in a set of predictions, without considering their direction. It is the average over the verification sample of the absolute

differences between prediction and actual observation. The mathematical representation of MAE is given by:

$$\text{MAE} = \frac{1}{K \times T} \sum_{k=1}^{K} \sum_{t=1}^{T} |x_t^k - \hat{x}_t^k|, \qquad (46)$$

where $K$ represents the number of time series (or variables) and $T$ represents the number of time steps.

**Continuous Ranked Probability Score (CRPS)** The Continuous Ranked Probability Score (CRPS) (Matheson & Winkler, 1976) quantifies the agreement between a cumulative distribution function (CDF) $F$ and an observation $x$, represented as:

$$\text{CRPS} = \int_{\mathbb{R}} (F(z) - \mathbb{I}\{x \leq z\})^2 dz, \qquad (47)$$

where $\mathbb{I}\{x \leq z\}$ denotes the indicator function, equating to one if $x \leq z$ and zero otherwise. Being a proper scoring function, CRPS reaches its minimum when the predictive distribution $F$ coincides with the data distribution. When using the empirical CDF of $F$, denoted as $\hat{F}(z) = \frac{1}{n} \sum_{i=1}^{n} \mathbb{I}\{X_i \leq z\}$, where $n$ represents the number of samples $X_i \sim F$, CRPS can be precisely calculated from the simulated samples of the conditional distribution $p_\theta(\mathbf{x}_t|\mathbf{h}_t)$. In our practice, 100 samples are employed to estimate the empirical CDF.

### D.4. Implementation Details

We utilize the TFB (Qiu et al., 2024) and ProbTS (Zhang et al., 2024a)code repository for unified evaluation, with all baseline results also derived from TFB. Following the settings in TFB (Qiu et al., 2024) and TSFM-Bench (Li et al., 2025), we do not apply the "Drop Last" trick to ensure a fair comparison. All experiments of KITE are conducted using PyTorch (Paszke et al., 2019) in Python 3.8 and executed on an NVIDIA Tesla-A800 GPU. The initial batch size is set to 128, with the flexibility to halve it (down to a minimum of 8) in case of an Out-Of-Memory (OOM) issue. For deterministic forecasting, we adopt the mean of multiple samples as the final prediction. To ensure reproducibility and facilitate experimentation, datasets and code are available at: https://github.com/decisionintelligence/KITE.

**More Analysis.** We provide supplementary evaluations, including (1) the Parameter Sensitivity Analysis of $\sigma_{\min}$, hidden dimension $d$, CFG guidance scale $\gamma$, and KGC attention layers $L$ (Appendix C.1), and (2) The Efficiency Analysis (Appendix C.2).

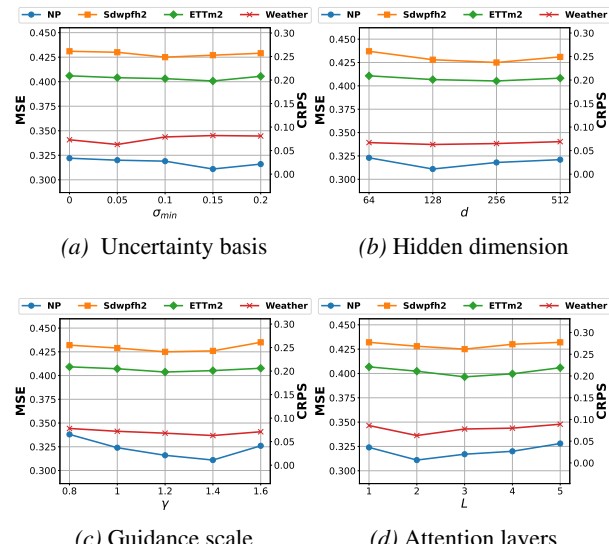

*(a)* Uncertainty basis    *(b)* Hidden dimension

*(c)* Guidance scale    *(d)* Attention layers

*Figure 3.* Parameter sensitivity analysis.

## E. More Analysis

### E.1. Parameter Sensitivity Analysis

We study the parameter sensitivity of KITE, including the uncertainty basis $\sigma_{\min}$ in *Uncertainty Estimator*, the the hidden dimension size $d$ of the model, the guidance scale $\gamma$ in *Classifier-Free Guidance*, and the number of attention layers $L$ in *Knowledge-Guided Conditioning*. We conducted experiments on both deterministic and probabilistic forecasting, reporting MSE and CRPS, respectively. Figure 3a shows the impact of $\sigma_{\min}$. In most cases, a small $K$ is insufficient for effective learning. We typically select $[0.05, 0.15]$ as the preferred option. Besides, Figure 3b illustrates the impact of $d$. Common choices for this value are 128 or 256. As illustrated in Figure 3c, the guidance scale $\gamma$ modulates the generation process: $\gamma < 1$ represents a weighted interpolation between conditional and unconditional modes, while $\gamma > 1$ steers the output away from the unconditional distribution (i.e., without covariates). Empirical results indicate that values of 1.2 or 1.4 are optimal in most cases, highlighting the forecasting task's strong dependency on covariates. Figure 3d indicates that deep layers are not required, thus avoiding substantial additional computational costs.

### E.2. Efficiency Analysis

We evaluate the model efficiency from three aspects: probabilistic forecasting performance (CRPS), inference time (sec/sample), and number of parameter memory (M). Figure 4 showcases a common scenario on Weather-L ($T = 96, F = 192$), which reflects the overall relative relationships on above-mentioned three aspects. KITE achieves the best forecasting performance among all baselines, while also securing the fastest inference speed specifically within

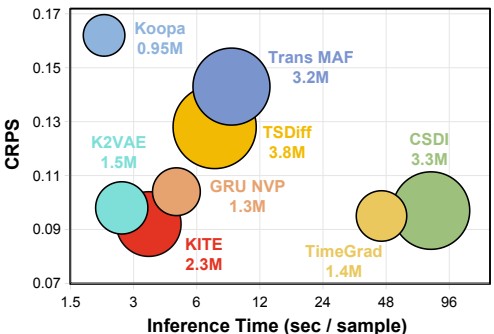

*Figure 4.* Model efficiency comparison. All the statistical data is obtained on the Weather-L ($T = 96, F = 192$). Lower values of CRPS indicate better performance.

channel-dependent models. This demonstrates that the explicit modeling guided by statistical knowledge imposes no additional computational burden; instead, it facilitates the accurate capture of relationships between endogenous and exogenous variables.

## F. Full Results

We present the comprehensive MSE and MAE results for KITE's deterministic forecasting in Tables 8 and 9, corresponding to settings with and without future exogenous variables, respectively. The evaluation spans 12 real-world datasets across both short-term and long-term prediction. For the Colbun and Raperl datasets, the short-term prediction uses a lookback window of 60 with a prediction horizon of 10, while the long-term prediction uses a lookback window of 180 with a prediction horizon of 30. For the other datasets, the short-term prediction uses a lookback window of 168 with a prediction horizon of 24, and the long-term prediction uses a lookback window of 720 with a prediction horizon of 360.

*Table 8.* Full results on deterministic forecasting with historical and future exogenous variables across 12 real-world datasets, where the inputs are $X_{endo}$, $X_{exo}$, and $Y_{exo}$. **Red**: the best, Blue: the 2nd best. Avg means the average results from two forecasting horizons.

| Models | Metrics | KITE (ours) mse | mae | TimeXer mse | mae | TFT mse | mae | TiDE mse | mae | DUET mse | mae | CrossLinear mse | mae | Amplifier mse | mae | TimeKAN mse | mae | xPatch mse | mae | PatchTST mse | mae | DLinear mse | mae |
|---|---|---|---|---|---|---|---|---|---|---|---|---|---|---|---|---|---|---|---|---|---|---|---|
| NP | 24 | 0.179 | 0.223 | 0.236 | 0.266 | 0.219 | 0.249 | 0.284 | 0.301 | 0.246 | 0.287 | 0.210 | 0.266 | 0.252 | 0.303 | 0.273 | 0.310 | 0.234 | 0.278 | 0.249 | 0.294 | 0.282 | 0.337 |
| | 360 | 0.471 | 0.422 | 0.600 | 0.475 | 0.539 | 0.501 | 0.601 | 0.498 | 0.576 | 0.528 | 0.531 | 0.508 | 0.587 | 0.534 | 0.538 | 0.529 | 0.523 | 0.461 | 0.530 | 0.498 | 0.601 | 0.552 |
| | Avg | 0.325 | 0.323 | 0.418 | 0.370 | 0.379 | 0.375 | 0.443 | 0.399 | 0.411 | 0.407 | 0.370 | 0.387 | 0.419 | 0.418 | 0.406 | 0.419 | 0.378 | 0.370 | 0.390 | 0.396 | 0.442 | 0.444 |
| PJM | 24 | 0.056 | 0.143 | 0.075 | 0.166 | 0.095 | 0.195 | 0.106 | 0.214 | 0.072 | 0.166 | 0.088 | 0.191 | 0.096 | 0.208 | 0.115 | 0.244 | 0.077 | 0.168 | 0.116 | 0.239 | 0.096 | 0.208 |
| | 360 | 0.135 | 0.214 | 0.140 | 0.231 | 0.133 | 0.219 | 0.177 | 0.279 | 0.131 | 0.228 | 0.135 | 0.254 | 0.177 | 0.285 | 0.162 | 0.281 | 0.131 | 0.220 | 0.149 | 0.279 | 0.164 | 0.278 |
| | Avg | 0.096 | 0.178 | 0.108 | 0.199 | 0.114 | 0.207 | 0.141 | 0.246 | 0.102 | 0.197 | 0.112 | 0.223 | 0.137 | 0.246 | 0.139 | 0.263 | 0.104 | 0.194 | 0.133 | 0.259 | 0.130 | 0.243 |
| BE | 24 | 0.348 | 0.240 | 0.392 | 0.253 | 0.426 | 0.272 | 0.426 | 0.285 | 0.432 | 0.272 | 0.391 | 0.259 | 0.471 | 0.339 | 0.451 | 0.319 | 0.411 | 0.256 | 0.452 | 0.326 | 0.451 | 0.310 |
| | 360 | 0.507 | 0.332 | 0.512 | 0.327 | 0.482 | 0.310 | 0.571 | 0.364 | 0.597 | 0.436 | 0.568 | 0.416 | 0.646 | 0.487 | 0.645 | 0.495 | 0.579 | 0.410 | 0.702 | 0.538 | 0.600 | 0.442 |
| | Avg | 0.427 | 0.286 | 0.452 | 0.290 | 0.454 | 0.291 | 0.498 | 0.325 | 0.514 | 0.354 | 0.479 | 0.338 | 0.558 | 0.413 | 0.548 | 0.407 | 0.495 | 0.333 | 0.577 | 0.432 | 0.526 | 0.376 |
| FR | 24 | 0.311 | 0.178 | 0.366 | 0.208 | 0.543 | 0.253 | 0.418 | 0.255 | 0.384 | 0.251 | 0.390 | 0.226 | 0.459 | 0.348 | 0.454 | 0.296 | 0.424 | 0.235 | 0.518 | 0.368 | 0.422 | 0.265 |
| | 360 | 0.462 | 0.271 | 0.489 | 0.273 | 0.465 | 0.261 | 0.551 | 0.308 | 0.607 | 0.403 | 0.575 | 0.370 | 0.648 | 0.468 | 0.641 | 0.452 | 0.556 | 0.350 | 0.658 | 0.452 | 0.572 | 0.362 |
| | Avg | 0.387 | 0.225 | 0.427 | 0.240 | 0.504 | 0.257 | 0.485 | 0.281 | 0.495 | 0.327 | 0.482 | 0.298 | 0.553 | 0.408 | 0.547 | 0.374 | 0.490 | 0.293 | 0.588 | 0.410 | 0.497 | 0.314 |
| DE | 24 | 0.262 | 0.327 | 0.339 | 0.362 | 0.380 | 0.383 | 0.367 | 0.383 | 0.376 | 0.378 | 0.387 | 0.396 | 0.394 | 0.407 | 0.399 | 0.412 | 0.399 | 0.394 | 0.413 | 0.412 | 0.423 | 0.426 |
| | 360 | 0.437 | 0.413 | 0.610 | 0.474 | 0.599 | 0.509 | 0.630 | 0.511 | 0.589 | 0.482 | 0.583 | 0.507 | 0.551 | 0.474 | 0.547 | 0.479 | 0.550 | 0.472 | 0.589 | 0.498 | 0.573 | 0.487 |
| | Avg | 0.350 | 0.370 | 0.475 | 0.418 | 0.489 | 0.446 | 0.498 | 0.447 | 0.482 | 0.430 | 0.485 | 0.452 | 0.473 | 0.441 | 0.473 | 0.446 | 0.474 | 0.433 | 0.501 | 0.455 | 0.498 | 0.457 |
| Energy | 24 | 0.071 | 0.208 | 0.122 | 0.273 | 0.093 | 0.235 | 0.103 | 0.248 | 0.117 | 0.283 | 0.241 | 0.418 | 0.138 | 0.306 | 0.135 | 0.298 | 0.092 | 0.240 | 0.239 | 0.390 | 0.085 | 0.230 |
| | 360 | 0.151 | 0.305 | 0.204 | 0.357 | 0.167 | 0.331 | 0.202 | 0.355 | 0.288 | 0.452 | 0.237 | 0.385 | 0.328 | 0.472 | 0.302 | 0.464 | 0.204 | 0.367 | 0.214 | 0.363 | 0.263 | 0.416 |
| | Avg | 0.111 | 0.257 | 0.163 | 0.315 | 0.130 | 0.283 | 0.152 | 0.301 | 0.202 | 0.367 | 0.239 | 0.401 | 0.233 | 0.389 | 0.218 | 0.381 | 0.148 | 0.303 | 0.226 | 0.377 | 0.174 | 0.323 |
| Sdwpfm1 | 24 | 0.384 | 0.410 | 0.558 | 0.533 | 0.366 | 0.421 | 0.474 | 0.488 | 0.551 | 0.564 | 0.355 | 0.473 | 0.346 | 0.445 | 0.418 | 0.503 | 0.669 | 0.580 | 0.374 | 0.454 | 0.417 | 0.501 |
| | 360 | 0.450 | 0.491 | 0.845 | 0.684 | 0.597 | 0.528 | 0.492 | 0.526 | 0.646 | 0.577 | 0.497 | 0.532 | 0.510 | 0.535 | 0.476 | 0.566 | 0.592 | 0.556 | 0.497 | 0.550 | 0.480 | 0.562 |
| | Avg | 0.417 | 0.451 | 0.702 | 0.609 | 0.481 | 0.475 | 0.483 | 0.507 | 0.599 | 0.571 | 0.426 | 0.502 | 0.428 | 0.490 | 0.447 | 0.534 | 0.630 | 0.568 | 0.435 | 0.502 | 0.449 | 0.531 |
| Sdwpfm2 | 24 | 0.443 | 0.450 | 0.627 | 0.570 | 0.411 | 0.458 | 0.461 | 0.492 | 0.445 | 0.452 | 0.477 | 0.536 | 0.394 | 0.462 | 0.474 | 0.538 | 0.508 | 0.478 | 0.418 | 0.492 | 0.459 | 0.530 |
| | 360 | 0.506 | 0.515 | 0.978 | 0.736 | 0.541 | 0.519 | 0.511 | 0.540 | 0.584 | 0.528 | 0.589 | 0.611 | 0.587 | 0.563 | 0.520 | 0.589 | 0.523 | 0.513 | 0.602 | 0.603 | 0.589 | 0.630 |
| | Avg | 0.475 | 0.483 | 0.802 | 0.653 | 0.476 | 0.489 | 0.486 | 0.516 | 0.514 | 0.490 | 0.533 | 0.574 | 0.490 | 0.512 | 0.497 | 0.564 | 0.516 | 0.495 | 0.510 | 0.547 | 0.524 | 0.580 |
| Sdwpfh1 | 24 | 0.424 | 0.464 | 0.651 | 0.587 | 0.401 | 0.460 | 0.434 | 0.489 | 0.527 | 0.513 | 0.548 | 0.585 | 0.576 | 0.627 | 0.511 | 0.582 | 0.524 | 0.505 | 0.422 | 0.505 | 0.557 | 0.613 |
| | 360 | 0.457 | 0.517 | 0.841 | 0.700 | 0.557 | 0.523 | 0.472 | 0.527 | 0.551 | 0.519 | 0.566 | 0.601 | 0.497 | 0.569 | 0.643 | 0.694 | 0.472 | 0.504 | 0.513 | 0.548 | 0.600 | 0.635 |
| | Avg | 0.441 | 0.491 | 0.746 | 0.643 | 0.479 | 0.492 | 0.453 | 0.508 | 0.539 | 0.516 | 0.557 | 0.593 | 0.536 | 0.598 | 0.577 | 0.638 | 0.498 | 0.505 | 0.468 | 0.526 | 0.579 | 0.624 |
| Sdwpfh2 | 24 | 0.477 | 0.498 | 0.820 | 0.677 | 0.474 | 0.493 | 0.579 | 0.553 | 0.629 | 0.563 | 0.468 | 0.540 | 0.473 | 0.533 | 0.580 | 0.614 | 0.640 | 0.554 | 0.457 | 0.527 | 0.633 | 0.653 |
| | 360 | 0.524 | 0.555 | 0.962 | 0.761 | 0.657 | 0.549 | 0.619 | 0.614 | 0.665 | 0.569 | 0.608 | 0.609 | 0.569 | 0.628 | 0.713 | 0.729 | 0.579 | 0.547 | 0.641 | 0.632 | 0.704 | 0.674 |
| | Avg | 0.500 | 0.526 | 0.891 | 0.719 | 0.566 | 0.521 | 0.599 | 0.584 | 0.647 | 0.566 | 0.538 | 0.575 | 0.521 | 0.581 | 0.646 | 0.671 | 0.610 | 0.551 | 0.549 | 0.580 | 0.669 | 0.664 |
| Colbun | 10 | 0.055 | 0.101 | 0.113 | 0.172 | 0.092 | 0.135 | 0.089 | 0.131 | 0.089 | 0.134 | 0.071 | 0.102 | 0.071 | 0.121 | 0.061 | 0.101 | 0.090 | 0.104 | 0.062 | 0.122 | 0.069 | 0.151 |
| | 30 | 0.121 | 0.243 | 0.176 | 0.299 | 0.383 | 0.460 | 0.240 | 0.322 | 0.307 | 0.397 | 0.182 | 0.288 | 0.275 | 0.370 | 0.195 | 0.249 | 0.217 | 0.315 | 0.417 | 0.496 | 0.245 | 0.285 |
| | Avg | 0.088 | 0.172 | 0.144 | 0.235 | 0.237 | 0.297 | 0.164 | 0.227 | 0.198 | 0.266 | 0.127 | 0.195 | 0.173 | 0.245 | 0.128 | 0.175 | 0.153 | 0.210 | 0.239 | 0.309 | 0.157 | 0.218 |
| Rapel | 10 | 0.192 | 0.231 | 0.301 | 0.308 | 0.201 | 0.253 | 0.228 | 0.271 | 0.174 | 0.219 | 0.163 | 0.209 | 0.181 | 0.227 | 0.174 | 0.231 | 0.161 | 0.216 | 0.163 | 0.218 | 0.184 | 0.239 |
| | 30 | 0.296 | 0.358 | 0.387 | 0.416 | 0.409 | 0.414 | 0.411 | 0.432 | 0.365 | 0.432 | 0.340 | 0.417 | 0.333 | 0.416 | 0.325 | 0.390 | 0.505 | 0.546 | 0.374 | 0.445 | 0.316 | 0.410 |
| | Avg | 0.244 | 0.294 | 0.344 | 0.362 | 0.305 | 0.334 | 0.320 | 0.352 | 0.269 | 0.326 | 0.252 | 0.313 | 0.257 | 0.322 | 0.249 | 0.310 | 0.333 | 0.381 | 0.269 | 0.332 | 0.250 | 0.324 |
| 1st Count | | 17 | 16 | 0 | 0 | 2 | 4 | 0 | 0 | 1 | 0 | 0 | 1 | 2 | 0 | 0 | 0 | 1 | 3 | 1 | 0 | 0 | 0 |

*Table 9.* Full results on deterministic forecasting with historical exogenous variables across 12 real-world datasets, where the inputs are $X_{\text{endo}}$ and $X_{\text{exo}}$. **Red**: the best, Blue: the 2nd best. Avg means the average results from two forecasting horizons.

| Models | Metrics | KITE (ours) mse | mae | TimeXer mse | mae | TFT mse | mae | TiDE mse | mae | DUET mse | mae | CrossLinear mse | mae | Amplifier mse | mae | TimeKAN mse | mae | xPatch mse | mae | PatchTST mse | mae | DLinear mse | mae |
|---|---|---|---|---|---|---|---|---|---|---|---|---|---|---|---|---|---|---|---|---|---|---|---|
| NP | 24 | **0.220** | **0.257** | 0.270 | 0.292 | 0.330 | 0.321 | 0.297 | 0.309 | 0.262 | 0.271 | 0.246 | 0.282 | 0.302 | 0.321 | 0.336 | 0.344 | 0.283 | 0.296 | 0.283 | 0.296 | 0.294 | 0.308 |
| | 360 | **0.590** | **0.472** | 0.609 | 0.474 | 0.964 | 0.655 | 0.638 | 0.523 | 0.626 | 0.494 | 0.655 | 0.505 | 0.738 | 0.574 | 0.632 | 0.527 | 0.630 | 0.505 | 0.630 | 0.505 | 0.647 | 0.537 |
| | Avg | **0.405** | **0.365** | 0.440 | 0.383 | 0.647 | 0.488 | 0.468 | 0.416 | 0.444 | 0.383 | 0.451 | 0.394 | 0.520 | 0.448 | 0.484 | 0.436 | 0.457 | 0.401 | 0.457 | 0.401 | 0.471 | 0.423 |
| PJM | 24 | **0.068** | **0.160** | 0.096 | 0.191 | 0.129 | 0.231 | 0.105 | 0.213 | 0.082 | 0.178 | 0.096 | 0.198 | 0.093 | 0.195 | 0.130 | 0.236 | 0.106 | 0.212 | 0.106 | 0.212 | 0.109 | 0.215 |
| | 360 | **0.185** | **0.260** | 0.186 | 0.267 | 0.270 | 0.310 | 0.210 | 0.292 | 0.198 | 0.273 | 0.197 | 0.283 | 0.210 | 0.294 | 0.219 | 0.304 | 0.189 | 0.273 | 0.189 | 0.273 | 0.199 | 0.285 |
| | Avg | **0.127** | **0.210** | 0.141 | 0.229 | 0.200 | 0.271 | 0.158 | 0.253 | 0.140 | 0.226 | 0.147 | 0.241 | 0.152 | 0.245 | 0.175 | 0.270 | 0.148 | 0.243 | 0.148 | 0.243 | 0.154 | 0.250 |
| BE | 24 | **0.355** | 0.255 | 0.392 | **0.253** | 0.400 | 0.267 | 0.492 | 0.322 | 0.376 | 0.253 | 0.389 | 0.253 | 0.417 | 0.283 | 0.424 | 0.280 | 0.405 | 0.264 | 0.405 | 0.264 | 0.453 | 0.308 |
| | 360 | **0.550** | 0.349 | 0.563 | 0.349 | 0.727 | 0.442 | 0.602 | 0.375 | 0.571 | 0.357 | 0.564 | **0.346** | 0.588 | 0.367 | 0.552 | 0.350 | 0.564 | 0.354 | 0.564 | 0.354 | 0.645 | 0.454 |
| | Avg | **0.453** | 0.302 | 0.478 | 0.301 | 0.564 | 0.355 | 0.547 | 0.349 | 0.474 | 0.305 | 0.477 | **0.300** | 0.503 | 0.325 | 0.488 | 0.315 | 0.485 | 0.309 | 0.485 | 0.309 | 0.549 | 0.381 |
| FR | 24 | **0.337** | 0.197 | 0.366 | **0.195** | 0.445 | 0.231 | 0.415 | 0.254 | 0.359 | 0.208 | 0.397 | 0.204 | 0.429 | 0.250 | 0.448 | 0.250 | 0.397 | 0.223 | 0.397 | 0.223 | 0.412 | 0.253 |
| | 360 | **0.521** | **0.297** | 0.542 | 0.300 | 0.624 | 0.345 | 0.574 | 0.325 | 0.577 | 0.317 | 0.555 | 0.310 | 0.559 | 0.322 | 0.534 | 0.303 | 0.542 | 0.306 | 0.542 | 0.306 | 0.576 | 0.352 |
| | Avg | **0.429** | **0.247** | 0.454 | 0.248 | 0.535 | 0.288 | 0.495 | 0.290 | 0.468 | 0.263 | 0.476 | 0.257 | 0.494 | 0.286 | 0.491 | 0.277 | 0.470 | 0.265 | 0.470 | 0.264 | 0.494 | 0.302 |
| DE | 24 | **0.418** | **0.402** | 0.501 | 0.445 | 0.576 | 0.452 | 0.465 | 0.433 | 0.424 | 0.403 | 0.438 | 0.423 | 0.493 | 0.455 | 0.504 | 0.462 | 0.503 | 0.450 | 0.503 | 0.450 | 0.511 | 0.458 |
| | 360 | **0.783** | **0.566** | 0.818 | 0.568 | 0.792 | 0.578 | 0.823 | 0.604 | 0.895 | 0.623 | 0.832 | 0.592 | 0.930 | 0.642 | 0.834 | 0.590 | 0.889 | 0.604 | 0.889 | 0.604 | 0.868 | 0.600 |
| | Avg | **0.601** | **0.484** | 0.660 | 0.507 | 0.684 | 0.515 | 0.644 | 0.519 | 0.660 | 0.513 | 0.635 | 0.508 | 0.712 | 0.549 | 0.669 | 0.526 | 0.696 | 0.527 | 0.696 | 0.527 | 0.690 | 0.529 |
| Energy | 24 | 0.103 | 0.247 | 0.138 | 0.293 | 0.352 | 0.463 | 0.117 | 0.265 | 0.108 | 0.254 | **0.102** | **0.246** | 0.108 | 0.254 | 0.109 | 0.254 | 0.108 | 0.254 | 0.108 | 0.254 | 0.106 | 0.248 |
| | 360 | 0.188 | 0.340 | 0.206 | 0.360 | 0.399 | 0.497 | 0.207 | 0.358 | 0.211 | 0.362 | 0.299 | 0.425 | 0.251 | 0.400 | **0.186** | **0.338** | 0.298 | 0.428 | 0.298 | 0.428 | 0.214 | 0.367 |
| | Avg | **0.146** | **0.294** | 0.172 | 0.327 | 0.376 | 0.480 | 0.162 | 0.312 | 0.160 | 0.308 | 0.201 | 0.336 | 0.180 | 0.327 | 0.148 | 0.296 | 0.203 | 0.341 | 0.203 | 0.341 | 0.160 | 0.307 |
| Sdwpfm1 | 24 | **0.534** | **0.485** | 0.558 | 0.533 | 0.596 | 0.531 | 0.561 | 0.540 | 0.550 | 0.495 | 0.549 | 0.553 | 0.579 | 0.557 | 0.598 | 0.579 | 0.557 | 0.557 | 0.557 | 0.557 | 0.547 | 0.589 |
| | 360 | 0.862 | **0.669** | 0.845 | 0.684 | 1.372 | 0.925 | 0.864 | 0.725 | 0.898 | 0.672 | 1.070 | 0.834 | 1.106 | 0.814 | 0.853 | 0.764 | 0.910 | 0.745 | 0.910 | 0.745 | **0.839** | 0.773 |
| | Avg | 0.698 | **0.577** | 0.702 | 0.609 | 0.984 | 0.728 | 0.713 | 0.633 | 0.724 | 0.584 | 0.810 | 0.694 | 0.843 | 0.686 | 0.726 | 0.672 | 0.734 | 0.651 | 0.733 | 0.651 | **0.693** | 0.681 |
| Sdwpfm2 | 24 | 0.623 | **0.527** | 0.627 | 0.570 | 0.749 | 0.657 | 0.636 | 0.582 | **0.612** | 0.537 | 0.635 | 0.586 | 0.669 | 0.600 | 0.684 | 0.621 | 0.648 | 0.610 | 0.648 | 0.610 | 0.631 | 0.631 |
| | 360 | 0.977 | **0.717** | 0.978 | 0.736 | 1.340 | 0.877 | 0.995 | 0.783 | 1.028 | 0.745 | **0.965** | 0.788 | 1.215 | 0.863 | 0.987 | 0.780 | 1.020 | 0.819 | 1.020 | 0.819 | 0.980 | 0.824 |
| | Avg | **0.800** | **0.622** | 0.803 | 0.653 | 1.045 | 0.767 | 0.816 | 0.683 | 0.820 | 0.641 | 0.800 | 0.687 | 0.942 | 0.732 | 0.836 | 0.701 | 0.834 | 0.715 | 0.834 | 0.714 | 0.805 | 0.728 |
| Sdwpfh1 | 24 | **0.650** | **0.570** | 0.651 | 0.587 | 0.754 | 0.670 | 0.719 | 0.641 | 0.673 | 0.575 | 0.687 | 0.652 | 1.111 | 0.816 | 0.755 | 0.679 | 0.669 | 0.629 | 0.669 | 0.629 | 0.736 | 0.721 |
| | 360 | 0.869 | **0.683** | 0.841 | 0.700 | 0.831 | 0.727 | 0.897 | 0.757 | 0.886 | 0.712 | 0.849 | 0.752 | 1.089 | 0.823 | 0.853 | 0.762 | 0.939 | 0.804 | 0.939 | 0.804 | **0.829** | 0.781 |
| | Avg | 0.760 | **0.627** | **0.746** | 0.644 | 0.793 | 0.699 | 0.808 | 0.699 | 0.780 | 0.644 | 0.768 | 0.702 | 1.100 | 0.820 | 0.804 | 0.721 | 0.804 | 0.717 | 0.804 | 0.717 | 0.783 | 0.751 |
| Sdwpfh2 | 24 | 0.801 | **0.632** | 0.820 | 0.677 | 0.811 | 0.714 | 0.783 | 0.674 | 0.975 | 0.694 | 0.810 | 0.718 | 0.834 | 0.691 | 0.892 | 0.740 | **0.769** | 0.680 | 0.769 | 0.680 | 0.838 | 0.759 |
| | 360 | 0.958 | **0.749** | 0.962 | 0.761 | 1.042 | 0.779 | 1.056 | 0.829 | 1.039 | 0.735 | 1.102 | 0.830 | 1.126 | 0.845 | 0.990 | 0.825 | 1.280 | 0.925 | 1.280 | 0.925 | 0.989 | 0.837 |
| | Avg | **0.880** | **0.691** | 0.891 | 0.719 | 0.927 | 0.747 | 0.920 | 0.752 | 1.007 | 0.715 | 0.956 | 0.774 | 0.980 | 0.768 | 0.941 | 0.783 | 1.025 | 0.803 | 1.025 | 0.802 | 0.913 | 0.798 |
| Colbun | 10 | 0.069 | **0.098** | 0.084 | 0.146 | 0.449 | 0.279 | 0.095 | 0.140 | 0.073 | 0.106 | **0.068** | 0.101 | 0.077 | 0.120 | 0.091 | 0.129 | 0.083 | 0.125 | 0.083 | 0.125 | 0.077 | 0.135 |
| | 30 | 0.203 | **0.287** | 0.181 | 0.293 | 0.663 | 0.494 | 0.282 | 0.341 | 0.221 | 0.290 | 0.190 | 0.289 | 0.227 | 0.301 | 0.311 | 0.377 | 0.217 | 0.316 | 0.217 | 0.316 | 0.318 | 0.359 |
| | Avg | 0.136 | **0.193** | 0.133 | 0.220 | 0.556 | 0.387 | 0.189 | 0.241 | 0.147 | 0.198 | **0.129** | 0.195 | 0.152 | 0.211 | 0.201 | 0.253 | 0.150 | 0.221 | 0.150 | 0.221 | 0.198 | 0.247 |
| Rapel | 10 | 0.212 | 0.229 | 0.301 | 0.308 | 0.238 | 0.274 | 0.238 | 0.270 | 0.221 | 0.225 | **0.201** | **0.224** | 0.240 | 0.259 | 0.226 | 0.238 | 0.215 | 0.232 | 0.215 | 0.232 | 0.223 | 0.253 |
| | 30 | 0.392 | 0.382 | 0.387 | 0.416 | 0.377 | 0.394 | 0.409 | 0.444 | 0.387 | 0.386 | **0.278** | **0.374** | 0.433 | 0.436 | 0.458 | 0.437 | 0.436 | 0.436 | 0.436 | 0.436 | 0.414 | 0.423 |
| | Avg | 0.302 | 0.306 | 0.344 | 0.362 | 0.308 | 0.334 | 0.324 | 0.357 | 0.304 | 0.306 | **0.240** | **0.299** | 0.337 | 0.348 | 0.342 | 0.338 | 0.326 | 0.334 | 0.325 | 0.334 | 0.319 | 0.338 |
| 1st Count | | **13** | **16** | 1 | 2 | 0 | 0 | 0 | 0 | 1 | 1 | 5 | 4 | 0 | 0 | 1 | 1 | 1 | 0 | 0 | 0 | 2 | 0 |

