# OpenReview forum: "KITE: Knowledge-Guided Probabilistic Modeling for Time Series Forecasting with Exogenous Variables"
_ICML.cc/2026/Conference — ICML 2026 regular_

### Official Review · Reviewer_SPAD · 2026-03-10

**Soundness:** 2
**Presentation:** 3
**Significance:** 3
**Originality:** 3
**Overall Recommendation:** 4
**Confidence:** 4

**Summary:**

This paper proposes KITE, a probabilistic time series forecasting framework that incorporates exogenous variables via conditional flow matching. The authors identify two key bottlenecks in existing probabilistic forecasting approaches: (1) the topological mismatch between simple Gaussian priors and the complex target distribution in covariate-conditioned settings, and (2) the tendency of purely data-driven covariate modeling to overfit spurious correlations, which becomes particularly damaging when errors propagate through iterative generative sampling. To address these, KITE introduces a History-Conditional Manifold (HCM) that constructs a learnable, history-informed source distribution to replace the standard isotropic Gaussian prior, and a Knowledge-Guided Conditioning (KGC) module that injects statistical priors (e.g., Pearson correlation, Granger causality) into a bilinear attention mechanism to modulate the cross-variable dependency structure. Classifier-Free Guidance (CFG) is additionally incorporated to allow flexible interpolation between covariate-agnostic and covariate-conditioned generation. The authors provide both theoretical analyses (generalization bound and trajectory regularity) and empirical evaluations across 12 deterministic and 8 probabilistic forecasting benchmarks.

**Compliance With Llm Reviewing Policy:**

Affirmed.

**Final Justification:**

The additional results help to address my concerns and I have raised my score accordingly.

**Key Questions For Authors:**

See the Weakness.If the authors can resolve these questions during the rebuttal, I am highly open to raising my score.

**Strengths And Weaknesses:**

Strengths

1.The combination of a learnable history-conditioned source distribution with statistical-prior-guided bilinear attention is not a trivial extension of prior work. TSFlow (Kollovieh et al., 2025) uses GP priors but with fixed, non-learnable rules; KITE's HCM is adaptive and trained end-to-end. The bilinear attention formulation in KGC (Eq. 9–12), which interpolates between a data-driven subspace and a prior-injected subspace conditioned on per-pair statistical strength, is a reasonably clever mechanism that goes beyond naive concatenation of prior weights. The gating via $log⁡(S+δ)$ to suppress attention from near-zero-correlation pairs is a sensible detail.

2.The empirical scope is broad: 12 datasets for deterministic forecasting and 8 for probabilistic, covering electricity price markets, wind power, hydropower, weather, and standard ETT/Traffic/Exchange benchmarks. The ablation study in Table 1 is structured cleanly and isolates each of the three components (HCM, KGC, CFG). The statistical knowledge analysis in Table 4, which contrasts Pearson vs. Granger causality in different conditioning stages (historical vs. future covariates), is a thoughtful addition that adds interpretive value beyond the main results.

3.The paper is generally well-written. Figure 2 communicates the architecture reasonably clearly. Algorithm 1 is helpful for reproducibility. The authors have provided code at an anonymous repository.

4.The specific problem of probabilistic forecasting with exogenous variables (as opposed to multivariate forecasting where all variables are treated symmetrically) is indeed underserved in the literature. The framing of Figure 1 (three paradigms) is useful for positioning the work.

Weaknesses

1. The theoretical results (Theorems 4.1 and 4.2) are weakly formal. Theorem 4.1 asserts a generalization bound of the form $$E_{\text{gen}} \leq K \cdot \Psi\left(W_2(...)\right) \cdot \mathcal{R}_n + O(n^{-1/2})$$

where $Ψ$ is merely described as "a monotonically increasing function." This is not a concrete bound — any bound is trivially of this form for some choice of $Ψ$. The proof invokes the statement that "the Rademacher complexity of neural networks scales linearly with the output norm bound," citing Bartlett et al. (2017), but this result applies to margin-based spectral norm bounds for classifiers, not directly to regression with flow matching objectives. The connection between $W_2$ distance and the Rademacher complexity of the velocity estimator class is asserted rather than derived. Similarly, Theorem 4.2 states
$$L_{HCM} ≪ L_{Gauss} ⟹ W2²(P_{HCM},P_t) ≪ W2²(N,P_t)$$

but "≪" is used informally throughout and the implication is stated as intuitively plausible rather than formally derived from the Benamou–Brenier framework with concrete regularity conditions. These proofs read more as heuristic justifications than rigorous theoretical contributions.

2.The manifold parameterization in HCM is underdetermined. The source distribution is parameterized as $\mathcal{N}(\mu_{\text{hist}},\, \sigma^2_{\text{hist}} \cdot \Sigma)$, where $Σ$ is constructed via a low-rank manifold basis (Eq. 6). However, $Σ$ as defined in Eq. 6 is a vector in $R^{N×F}$, not a covariance matrix in $R^{(N×F)×(N×F)}$. The paper uses $Σ$ as if it defines an anisotropic covariance, but the actual sampling procedure is unclear: does it sample element-wise with per-element variances, or is there a proper low-rank covariance structure being modeled? This is a non-trivial distinction that affects the geometric argument central to HCM's motivation.

3.The Coverage Constraint (Eq. 7) is a single-sample point approximation.$L_{\text{CC}}$​ uses $|Y_{\text{endo}} - \text{detach}(\mu_{\text{hist}})|^2$ as a proxy for variance calibration. This is essentially fitting a Gaussian log-likelihood around the barycenter prediction. While this encourages the support of the source to cover the target, it does not formally guarantee that $W_2(P_{HCM},P_t)$ is small, because it only constrains the mean and scalar variance, not the full distributional geometry. The paper's claim that HCM "explicitly minimizes the transport cost" is an overstatement relative to what $ \mathcal{L}_{\text{CC}} $​ actually optimizes.

4.The bilinear attention formulation has a potential numerical issue. In Eq. 12, the term $\log(S + \delta)$ is added directly to the attention logits. Since $S$ contains Pearson correlations in [−1,1], values of $S<0$ make $S + δ$ negative for small $δ$, causing log to be
undefined. The paper says $δ$ is a "fixed small constant" but does not specify how it handles negative correlations or whether $S$ is clipped/rectified before this operation.

5.The exogenous variable protocol for probabilistic forecasting benchmarks is non-standard and potentially problematic. For probabilistic forecasting, the authors use standard multivariate datasets (ETT, Weather, Traffic, etc.) and artificially designate the last dimension as the endogenous target, treating all remaining dimensions as exogenous. This creates a covariate setup that introduces significant information leakage — particularly in the Traffic dataset (861 sensors, with 860 as covariates) and the Electricity dataset (321 clients, 320 as covariates). The authors compare KITE (Covariate Modeling with 860 covariates) against K2VAE and TSFlow (Univariate Modeling with no covariates). This is not a controlled comparison: KITE has access to massively more information. The improvement over univariate baselines in this setting does not demonstrate superior modeling of spurious correlations; it primarily reflects information advantage. At minimum, there should be a comparison where multivariate baselines also have access to the same covariate information as KITE.

6.For deterministic forecasting, the modification to non-native baselines (Algorithm 4) may not be the strongest possible adaptation. The paper adapts baselines without native covariate support using a simple MLP concatenation of the base model's output with $Y_{\text{exo}}$​. While the authors note this as a fair procedure, more capable adaptations exist (e.g., attention-based cross-conditioning). This risks systematically underestimating strong baselines.

7.The ablation study uses only 2 deterministic datasets (NP, Sdwpfh2) and 2 probabilistic (ETTm2, Weather). Given that the paper evaluates across 20 datasets total, it would be more convincing to show that the module ablation conclusions hold consistently across the full range of dataset types and sizes.

8.The reported 19.5% MSE reduction over the second-best baseline is averaged across all 12 deterministic datasets, but gains are unevenly distributed. Inspection of Table 7 shows that KITE's advantage is heavily concentrated in electricity price forecasting (NP, PJM, DE) and the Sdwpf wind datasets, while on datasets like Colbun and Rapel the margins are much smaller and occasionally KITE is not the best. A per-dataset breakdown of where gains originate would help readers assess generalizability.

9.Table 5 contains a typo: "Exhange" instead of "Exchange." The "Weather" dataset description lists the endogenous variable as "CO2-Concentration" in Table 5, but the appendix text says "Wet Bulb factor" — these are inconsistent.

10.The CFG formulation in Algorithm 1 differs from Eq. 16. Eq. 16 defines
$$v̂_s = (1-γ)·v(Y_s,∅) + γ·v(Y_s,c)$$
whereas Algorithm 1 computes
$$v̂=(1+γ)⋅v_cond​−γ⋅v_uncond​⋅$$
These are algebraically equivalent only when the Eq. 16 coefficient is rewritten — the paper should make this correspondence explicit rather than leaving it as an implicit reparameterization.

11.The paper does not compare against ChronosX (Arango et al., 2025), which is a recent foundation-model-based approach for time series forecasting with exogenous variables. This is cited in the related work but absent from the baselines, without justification.

12.DAG (Qiu et al., 2025b) is cited in the dataset sourcing but not included as a baseline, despite being directly relevant as a causal network for time series forecasting with exogenous variables.

---

> ### Author Rebuttal · Authors · 2026-03-31
>
> We thank Reviewer SPAD for the detailed and insightful review.
>
> ## W1
> Thank you for pointing this out. Theorems 4.1 and 4.2 are meant to provide intuition for why HCM is preferable to a standard Gaussian prior, rather than serve as core theoretical contributions. We will restate them as defensible propositions. Due to space limits, the updated versions are included in our response to **Reviewer qUpg** (# Theoretical analysis); full derivations will be provided in the discussion phase following your response.
>
> ## W2
> Thank you for this important clarification. We agree that Eq. (6) is imprecise: $\Sigma$ is not a covariance matrix. In implementation, HCM samples a structured perturbation, $\delta=\alpha \frac{Mz}{|Mz|}+(1-\alpha)\epsilon$, and sets $Y_0=\mu_{\text{hist}}+\sigma_{\text{hist}}\delta$. Thus, HCM is an implicit history-conditioned source sampler, not an explicit Gaussian $\mathcal{N}(\mu_{\text{hist}},\sigma_{\text{hist}}^2\Sigma)$. It models neither element-wise variances nor an explicit low-rank covariance; the anisotropy comes from the low-rank subspace induced by $M$. This distinction is important, but it does **not change the geometric argument**, which concerns the induced source distribution $P_{\mathrm{HCM}}$ being closer to the target than standard Gaussian initialization, rather than a closed-form covariance parameterization. We will revise Eq. (6) and the surrounding text accordingly.
> ## W3
> We agree that Eq. (7) is a single-sample Gaussian-NLL-style surrogate with scalar variance, not a direct objective on $W_2(P_{\mathrm{HCM}},P_t)$ or full distributional geometry. Its purpose is to prevent variance collapse and encourage target coverage. Thus, “explicitly minimizes the transport cost” is too strong; we will revise this to say that HCM provides a history-aligned initialization that can reduce transport cost relative to a standard Gaussian prior in practice, and we will revise the related descriptions accordingly.
> ## W4
> We agree that Eq. 12 is incomplete as written. In implementation, we use $\log(|S|+\delta)$ rather than $\log(S+\delta)$, so the term is always well-defined and $S$ is treated as a nonnegative prior strength. We will revise Eq. 12 accordingly.
> ## W5
> We would like to clarify the following points:
> 1. Following existing work (TimeXer NeurIPS 2024,  Cross-Linear SIGKDD 2025) , we construct this setting by designating the last one dimension as the endogenous target and treating others as exogenous covariates. Importantly, this setting uses only historical covariates, not future-window covariates. Therefore, there is no future
> information leakage.
> 2. To support a fair comparison, we also provide multivariate baselines in Table 3, which use the same inputs as KITE. For further fairness, we equip univariate baselines with the Exogenous Mixer from TTM [1] to incorporate covariates; results at https://anonymous.4open.science/r/KITE-8783/Uni.png further validate KITE's effectiveness.
>
> [1] Tiny Time Mixers (TTMs): Fast Pre-trained Models for Enhanced Zero/Few-Shot Forecasting of Multivariate Time Series. NeurIPS 2024
> ## W6
> We clarify that the MLP-based fusion is not necessarily too simplistic; the Exogenous Mixer in TTM uses a similar mechanism. We additionally test future-covariate fusion based on CNN and cross-attention. Results are reported at https://anonymous.4open.science/r/KITE-8783/Fusion.png.
> ## W7
> To  address this concern, we conducted additional ablation studies; results are summarized at https://anonymous.4open.science/r/KITE-8783/Ablation.png.
> ## W8
> To address this concern, we summarize the per-dataset breakdown using each dataset’s own second-best baseline as reference: NP 17.76%, PJM 14.80%, BE 8.14%, FR 14.53%, DE 28.32%, Energy 12.96%, Sdwpfm1 4.80%, Sdwpfm2 8.44%, Sdwpfh1 14.62%, Sdwpfh2 20.35%, Colbun 20.04%, Rapel 13.00%.
>
>  This variance may stem from differing correlation stability: the standard deviation of Pearson correlations across time windows is lower for Sdwpfm1 (0.121), whose stable patterns are easier for baselines to capture, and higher for Colbun (0.486), where KITE better handles non-stationarity than baselines.
>
>  Moreover, the suboptimal cases in Table 7 are mainly in long-term forecasting. We attribute this to the limited expressive power of the MLP-based $f_{\phi}$ in Barycenter Mapping. As shown at https://anonymous.4open.science/r/KITE-8783/SSM.png, replacing the MLP with a more expressive state-space model (e.g., Mamba) further improves performance.
>
> ## W9, W10
> Thank you for pointing out these issues. We will correct "Exhange" to "Exchange," revise the appendix to consistently use CO2-Concentration as the Weather endogenous variable, and make Algorithm 1 consistent with Eq. 16.
> ## W11, W12
> We additionally include DAG and ChronosX as baselines and evaluate them under the setting with $X_{exo}$ and $Y_{exo}$. Results are reported at https://anonymous.4open.science/r/KITE-8783/DAG&ChronosX.png.

---

> > ### Author Rebuttal · Reviewer_SPAD · 2026-04-02
> >
> > Thank you for the thorough and well-organized rebuttal. While my initial review raised a number of concerns, I am pleased to find that the authors have addressed the majority of them satisfactorily.
> >
> > Regarding W1, I note that the revised Propositions 4.1 and 4.2 represent a meaningful improvement over the original theorems. I would ask the authors to ensure that in the final revision, the original Theorems 4.1 and 4.2 are either fully replaced by the new propositions or clearly distanced from being presented as core theoretical contributions. A brief clarification on how the new propositions relate to the original proof sketches would also be helpful.
> >
> > Given the substantial effort and quality of this rebuttal, I am happy to raise my score. I trust that the authors will incorporate these final clarifications in the discussion phase and the camera-ready version.

---

> > > ### Author Response · Authors · 2026-04-03
> > >
> > > We sincerely thank the Reviewer SPAD for the valuable suggestions. We commit to replacing the original Theorems 4.1 and 4.2 with the Propositions 4.1 and 4.2 in the main text of the revised version. Furthermore, we will conduct a careful  check to ensure there are no formal errors.
> > >
> > > Relation to the original proof sketches: The new propositions maintain the same underlying motivations as the original sketches, but state them in a more direct and verifiable form.
> > >
> > > Proposition 4.1 is the training-target-scale counterpart of the generalization-gap sketch originally presented in Theorem 4.1. Both characterize how HCM brings the source closer to the target, thereby simplifying the regression task.
> > >
> > > Proposition 4.2 is the local-path counterpart of the trajectory-regularity sketch originally in Theorem 4.2. Both characterize how HCM yields a better-conditioned transport path than standard Gaussian initialization.
> > >
> > > We provide the detailed proofs for Propositions 4.1 and 4.2 below.
> > >
> > > **Proof for Proposition 4.1.** By definition of the HCM source,
> > >
> > > $$
> > > Y_0^H=\mu(h)+\sigma(h)\delta,
> > > $$
> > >
> > > and therefore
> > >
> > > $$
> > > Y-Y_0^H=(Y-\mu(h))-\sigma(h)\delta.
> > > $$
> > >
> > > Applying the triangle inequality for the $L_1$ norm gives
> > >
> > > $$
> > > \Vert Y-Y_0^H\Vert_1 \le \Vert Y-\mu(h)\Vert_1+\sigma(h)\Vert\delta\Vert_1.
> > > $$
> > >
> > > Conditioning on $(h,c)$ and noting that $\sigma(h)$ is $(h,c)$-measurable, we obtain
> > >
> > > $$
> > > \mathbb{E}[\Vert Y-Y_0^H\Vert_1\mid h,c] \le \mathbb{E}[\Vert Y-\mu(h)\Vert_1\mid h,c] + \sigma(h)\mathbb{E}[\Vert\delta\Vert_1\mid h,c].
> > > $$
> > >
> > > Since $\delta$ is independent of $(h,c)$, we have
> > >
> > > $$
> > > \mathbb{E}[\Vert\delta\Vert_1\mid h,c]=\mathbb{E}[\Vert\delta\Vert_1]=C_H.
> > > $$
> > >
> > > Hence,
> > >
> > > $$
> > > \mathbb{E}[\Vert Y-Y_0^H\Vert_1\mid h,c] \le \mathbb{E}[\Vert Y-\mu(h)\Vert_1\mid h,c]+\sigma(h)C_H.
> > > \tag{1}
> > > $$
> > >
> > > For the Gaussian source, since
> > >
> > > $$
> > > Y_0^G=\xi,
> > > $$
> > >
> > > we have
> > >
> > > $$
> > > Y-Y_0^G=Y-\xi.
> > > $$
> > >
> > > By the reverse triangle inequality,
> > >
> > > $$
> > > \Vert Y-\xi\Vert_1 \ge \Vert Y\Vert_1-\Vert\xi\Vert_1.
> > > $$
> > >
> > > Conditioning on $(h,c)$ yields
> > >
> > > $$
> > > \mathbb{E}[\Vert Y-Y_0^G\Vert_1\mid h,c] \ge \mathbb{E}[\Vert Y\Vert_1\mid h,c] - \mathbb{E}[\Vert\xi\Vert_1\mid h,c].
> > > $$
> > >
> > > Since $\xi$ is independent of $(h,c)$,
> > >
> > > $$
> > > \mathbb{E}[\Vert\xi\Vert_1\mid h,c]=\mathbb{E}[\Vert\xi\Vert_1]=C_G.
> > > $$
> > >
> > > Therefore,
> > >
> > > $$
> > > \mathbb{E}[\Vert Y-Y_0^G\Vert_1\mid h,c] \ge \mathbb{E}[\Vert Y\Vert_1\mid h,c]-C_G.
> > > \tag{2}
> > > $$
> > >
> > > Now assume that
> > >
> > > $$
> > > \mathbb{E}[\Vert Y\Vert_1\mid h,c]-\mathbb{E}[\Vert Y-\mu(h)\Vert_1\mid h,c] > \sigma(h)C_H+C_G.
> > > $$
> > >
> > > Rearranging gives
> > >
> > > $$
> > > \mathbb{E}[\Vert Y-\mu(h)\Vert_1\mid h,c]+\sigma(h)C_H < \mathbb{E}[\Vert Y\Vert_1\mid h,c]-C_G.
> > > \tag{3}
> > > $$
> > >
> > > Combining (1), (2), and (3), we obtain
> > >
> > > $$
> > > \mathbb{E}[\Vert Y-Y_0^H\Vert_1\mid h,c] < \mathbb{E}[\Vert Y\Vert_1\mid h,c]-C_G \le \mathbb{E}[\Vert Y-Y_0^G\Vert_1\mid h,c].
> > > $$
> > >
> > > Thus,
> > >
> > > $$
> > > \mathbb{E}[\Vert Y-Y_0^H\Vert_1\mid h,c] < \mathbb{E}[\Vert Y-Y_0^G\Vert_1\mid h,c].
> > > $$
> > >
> > > This proves the proposition. $\square$
> > >
> > >
> > > **Proof for Proposition 4.2.** Recall that
> > >
> > > $$
> > > Y_s^H=sY+(1-s)Y_0^H,\qquad Y_0^H=\mu(h)+\sigma(h)\delta.
> > > $$
> > >
> > > Subtracting $\mu(h)$ from both sides gives
> > >
> > > $$
> > > Y_s^H-\mu(h)=s\bigl(Y-\mu(h)\bigr)+(1-s)\sigma(h)\delta.
> > > $$
> > >
> > > Hence, for every $s\in[0,1]$, the point $Y_s^H-\mu(h)$ lies on the line segment between $Y-\mu(h)$ and $\sigma(h)\delta$. By convexity of the $L_1$ norm,
> > >
> > > $$
> > > \Vert Y_s^H-\mu(h)\Vert_1 \le \max\bigl(\Vert Y-\mu(h)\Vert_1,\ \sigma(h)\Vert\delta\Vert_1\bigr) \le \Vert Y-\mu(h)\Vert_1+\sigma(h)\Vert\delta\Vert_1.
> > > $$
> > >
> > > Taking the supremum over $s\in[0,1]$ yields
> > >
> > > $$
> > > R_H = \sup_{s\in[0,1]}\Vert Y_s^H-\mu(h)\Vert_1 \le \Vert Y-\mu(h)\Vert_1+\sigma(h)\Vert\delta\Vert_1.
> > > $$
> > >
> > > Therefore,
> > >
> > > $$
> > > \mathbb E[R_H\mid h,c] \le \mathbb E[\Vert Y-\mu(h)\Vert_1\mid h,c]+\sigma(h)C_H.
> > > \tag{1}
> > > $$
> > >
> > > For the Gaussian path,
> > >
> > > $$
> > > Y_s^G=sY+(1-s)Y_0^G,\qquad Y_0^G=\xi\sim\mathcal N(0,I).
> > > $$
> > >
> > > Since $Y_1^G=Y$, we have
> > >
> > > $$
> > > R_G = \sup_{s\in[0,1]}\Vert Y_s^G\Vert_1 \ge \Vert Y_1^G\Vert_1 = \Vert Y\Vert_1.
> > > $$
> > >
> > > Hence,
> > >
> > > $$
> > > \mathbb E[R_G\mid h,c]\ge \mathbb E[\Vert Y\Vert_1\mid h,c].
> > > \tag{2}
> > > $$
> > >
> > > Now assume that the condition of Proposition 4.1 holds:
> > >
> > > $$
> > > \mathbb E[\Vert Y\Vert_1\mid h,c]-\mathbb E[\Vert Y-\mu(h)\Vert_1\mid h,c] > \sigma(h)C_H+C_G.
> > > $$
> > >
> > > Since $C_G\ge 0$, this implies
> > >
> > > $$
> > > \mathbb E[\Vert Y-\mu(h)\Vert_1\mid h,c]+\sigma(h)C_H < \mathbb E[\Vert Y\Vert_1\mid h,c].
> > > \tag{3}
> > > $$
> > >
> > > Combining (1), (2), and (3), we obtain
> > >
> > > $$
> > > \mathbb E[R_H\mid h,c] < \mathbb E[R_G\mid h,c].
> > > \tag{4}
> > > $$
> > >
> > > Under the stated local regularity condition, any path tube of radius $R$ admits the Jacobian upper bound $a+bR$. Therefore, the quantities
> > >
> > > $$
> > > \mathcal B_H:=a+bR_H,\qquad \mathcal B_G:=a+bR_G
> > > $$
> > >
> > > are valid path-local Jacobian budgets for the HCM and Gaussian paths, respectively. Since $b>0$, taking conditional expectation in (4) gives
> > >
> > > $$
> > > \mathbb E[\mathcal B_H\mid h,c] = a+b\,\mathbb E[R_H\mid h,c] < a+b\,\mathbb E[R_G\mid h,c] = \mathbb E[\mathcal B_G\mid h,c].
> > > $$
> > >
> > > Thus,
> > >
> > > $$
> > > \mathbb E[\mathcal B_H\mid h,c] < \mathbb E[\mathcal B_G\mid h,c].
> > > $$
> > >
> > > This proves the proposition. $\square$

---

### Official Review · Reviewer_iFgU · 2026-03-13

**Soundness:** 3
**Presentation:** 3
**Significance:** 3
**Originality:** 3
**Overall Recommendation:** 5
**Confidence:** 3

**Summary:**

This paper addresses the bottlenecks of topological disparity and spurious covariate correlation in probabilistic forecasting. A framework is proposed that uses 1) history-conditional manifold (HCM) as a history-aware source distribution and 2) knowledge-guided conditioning (KGC) to integrate statistical knowledge into the dynamic model. Experiments have been conducted to demonstrate the improved performance of the proposed method in probabilistic and deterministic forecasting.

**Compliance With Llm Reviewing Policy:**

Affirmed.

**Final Justification:**

All my points are addressed in the authors' reply. The discussion with other reviewers and me has improved the soundness of the paper, and I believe that the authors will include them in the final version of the paper. I'll raise my score accordingly.

**Key Questions For Authors:**

1. Regarding HCM, could the authors discuss the complexity of the sampling process? What is the complexity of the manifold basis $M$? Some symbols in Section 4.2 are not consistent. For example, is $\sigma$ in Eq 5 the same as $\sigma_{hist}$?
2. Section 4.3 states the use of statistical knowledge $S$ in the attention mechanism. In Table 4, it shows that some types of statistical knowledge lead to suboptimal forecasting. Could the authors further explain why KGC is sensitive to the choice of statistical knowledge?
3. Section 5.1.2: Among probabilistic baselines, apart from univariate vs multivariate, are there other differences, for example, the choices of prior or the covariate correlation, among comparison models?
4. In Tables 3-4, although the proposed method has a superior performance to other methods, the improvement varies across different datasets. Is it because of how well the history prior is learned in the model? Could the authors have a qualitative analysis of the prior?
5. It would be good to have additional ablations on the choices of hyperparameters, such as the baseline exploration width $\sigma_{min}$, the rank of the manifold basis $r$, and the mixing weight $\alpha$.

**Limitations:**

Yes

**Strengths And Weaknesses:**

Strengths
1. The problems of context-agnostic prior and spurious covariate correlation are important in probabilistic forecasting. The idea of utilizing flow matching framework is novel.
2. The HCM is efficient in adapting the history-aware source distribution to target data.
3. The KGC is robust to spurious correlations by incorporating statistical priors in the attention mechanism.
4. Empirical results show that the proposed method has superior forecasting performance compared to existing methods in probabilistic and deterministic setups.

Weaknesses
1. The computational complexity of HCM is increased given additional sampling steps and neural functions.
2. Different choices of statistical knowledge will cause suboptimal adaptation of the attention mechanism.
3. More details of the experiments need to be disclosed and elaborated.

---

> ### Author Rebuttal · Authors · 2026-03-31
>
> We sincerely thank Reviewer iFgu for the positive feedback and the constructive suggestions.
>
> ## Response to W1, Q1
> **1. Computational complexity of HCM:** HCM introduces only a one-shot initialization cost prior to the ODE solving. Under the linear barycenter mapping (Eq. 4), this overhead is bounded by $\mathcal{O}(NTF + NFr)$, where $N, T, F$, and $r$ denotes the number of endogenous variables, look-back length, prediction horizon, and manifold rank, respectively. It adds no per-step sampling cost. Instead, by providing a source distribution closer to the target, it reduces the transport burden and better conditions the N-step ODE integration. See the empirical efficiency comparison at https://anonymous.4open.science/r/KITE-8783/Efficiency.png; compared to the standard implementation, HCM does not introduce significant performance overhead.
>
> **2. Complexity of the manifold basis $M$:** The manifold basis is a low-rank matrix $M \in \mathbb{R}^{(N\times F)\times r}$. Both its parameter complexity and the projection cost are $\mathcal{O}(NFr)$. As with HCM, this is a one-time overhead incurred during the initialization of $Y_0$.
>
> **3. Notation Consistency:** Thank you for the correction. We will unify the notation to $\sigma_{hist}$ in Eq. 5 to avoid ambiguity and rigorously double-check all symbols throughout the manuscript.
>
> ## Response to W2, Q2
> We thank you for this insightful comment. We would like to clarify that the suboptimal cases (line 3) in Table 4 are intentionally included mismatch settings, rather than recommended choices.  Since KGC uses statistical knowledge to directly modulate the attention mechanism, mismatched knowledge can bias attention toward suboptimal interaction patterns.
>
> Specifically, Granger causality is more suitable for capturing dependencies from historical exogenous variables to future endogenous variables, as it reflects lagged predictive effects. In contrast, Pearson correlation is more suitable for capturing dependencies between future exogenous variables and future endogenous variables, since these variables are available within the forecast horizon and their relationship is more synchronous than lag-driven. As shown in Table 4, correctly applying these two types of statistical knowledge leads to improved performance (line 4).  We will clarify this point in the revised manuscript.
>
> ## Response to Q3
> Yes, beyond the univariate/multivariate distinction, the baselines fundamentally differ in three underlying assumptions:
>
> 1. **Generative Families:** Diffusion (TimeGrad, TSDiff, CSDI), Normalizing-Flow (GRU-NVP, Trans-MAF), Latent-Dynamics (K2VAE), Point-Forecasting with Gaussian Heads (PatchTST, Koopa, iTransformer), and Flow-Matching (TSFlow, **KITE**).
> 2. **Source / Prior Distributions:** Standard Gaussian noise (TimeGrad, TSDiff, CSDI), Gaussian Process priors (TSFlow), Gaussian heads (PatchTST, Koopa, iTransformer), versus our **Learnable History-Conditioned Source (KITE)**.
> 3. **Covariate Correlation Mechanisms:** Implicit data-driven interactions via Attention (iTransformer, CSDI) or RNN/CNN/Flows (TimeGrad, GRU-NVP, Trans-MAF), versus our **Explicit Knowledge-Guided Interaction (KITE)**.
>
> ## Response to Q4
> We completely agree that the varying improvements stem from the quality of the learned history prior, particularly the effectiveness of the **Uncertainty Estimator**.
>
> As a qualitative analysis, we compared ETTh1 (2.13% avg. improvement) and Traffic (11.94% avg. improvement). Empirically, the average learned uncertainty scale ($\sigma_{hist}$, Eq. 5) reaches **0.37 on ETTh1**, but only **0.17 on Traffic**. This directly indicates that ETTh1's learned prior inherently carries much higher uncertainty. A broader source distribution naturally increases the transport distance and ODE sampling difficulty, which perfectly explains its relatively smaller performance margin compared to the highly concentrated prior learned for Traffic.
>
> ## Response to Q5
> Thank you for the constructive suggestion. The baseline exploration width ($\sigma_{min}$) is already analyzed in Appendix D.1. We have further evaluated the manifold basis rank ($r$) and the initialization of mixing weight ($\alpha$), summarized in Figure 4 (https://anonymous.4open.science/r/KITE-8783/Hyperparameters.png).
>
> Based on these results, we recommend setting the rank $r \in \{4, 8\}$. Furthermore, since $\alpha$ is a learnable parameter, KITE shows a certain degree of robustness to its initialization; thus, we typically initialize it to 0.5 or 0.6. We will include these additional ablations and discussions in Appendix D.1.
>
> **Thanks for your sincere suggestions again! If you have any additional questions, we can have further discussions!**

---

> > ### Author Rebuttal · Reviewer_iFgU · 2026-04-02
> >
> > All my points are addressed in the authors' reply. The discussion with other reviewers and me has improved the soundness of the paper, and I believe that the authors will include them in the final version of the paper. I'll raise my score accordingly.

---

> > > ### Author Response · Authors · 2026-04-03
> > >
> > > We sincerely thank Reviewer iFgU for the positive feedback. We will incorporate these additional experiments and discussions into the revised version to further enhance the completeness of our paper. Thank you again for your insightful guidance and support throughout the review process!

---

### Official Review · Reviewer_qUpg · 2026-03-15

**Soundness:** 3
**Presentation:** 3
**Significance:** 3
**Originality:** 3
**Overall Recommendation:** 5
**Confidence:** 3

**Summary:**

This paper proposes an approach to improve flow matching generative modeling for time series forecasting by constructing a source distribution informed by observed historical time series data and statistical relationships within the multivariate time series, including Pearson correlation and Granger causality. The authors provide theoretical analysis suggesting that the manifold induced by the historical data leads to improved generalization error bounds. Experiments are conducted across multiple datasets, comparing the proposed method against strong baselines in both point forecasting and probabilistic forecasting tasks.

**Compliance With Llm Reviewing Policy:**

Affirmed.

**Key Questions For Authors:**

What are the main assumptions you made for your theoretical analysis?

**Limitations:**

No, the authors didn't talk about limitations. Please let us know the main limitations of your work in the rebuttal phase.

**Strengths And Weaknesses:**

The methodology appears sound and the empirical results are strong. However, the paper provides limited discussion of prior work that departs from the standard assumption of Gaussian noise as the source distribution. For example, multimodal frameworks such as Ref1 and Ref2 have explored related directions, and discussing these works would help readers better understand the distinctions and contributions of the proposed approach.

In addition, the use of the term knowledge-guided to describe guidance via statistical measures is somewhat problematic. This term is widely used in the literature to refer to approaches that incorporate scientific or mechanistic knowledge, such as differential equations or domain-specific relationships (see Ref3 and Ref4). Using it in the context of statistical guidance may lead to confusion, and it may be helpful to clarify the terminology or adopt a more precise term.

[Ref1] He, J., Yu, Q., Liu, Q. and Chen, L.C., 2025. Flowtok: Flowing seamlessly across text and image tokens. In Proceedings of the IEEE/CVF International Conference on Computer Vision (pp. 16629-16640).

[Ref2] Liu, Q., Yin, X., Yuille, A., Brown, A. and Singh, M., 2025. Flowing from words to pixels: A noise-free framework for cross-modality evolution. In Proceedings of the Computer Vision and Pattern Recognition Conference (pp. 2755-2765).

[Ref3] Karpatne, A., Jia, X. and Kumar, V., 2024. Knowledge-guided machine learning: Current trends and future prospects. arXiv preprint arXiv:2403.15989.

[Ref4] Liu, L., Zhou, W., Guan, K., Peng, B., Xu, S., Tang, J., Zhu, Q., Till, J., Jia, X., Jiang, C. and Wang, S., 2024. Knowledge-guided machine learning can improve carbon cycle quantification in agroecosystems. Nature communications, 15(1), p.357.

---

> ### Author Rebuttal · Authors · 2026-03-31
>
> We sincerely thank Reviewer qUpg for the helpful suggestions.
> ## Discussion of prior work
> [Ref1], [Ref2], and KITE all replace standard Gaussian noise with a context-aware source, but differ in two ways:
> 1. Their source is cross-modal for alignment, whereas HCM is history-conditioned for geometric shaping in exogenous forecasting.
> 2. For the stronger uncertainty and variability in time-series, we further introduce an **Uncertainty Estimator** and a **low-rank Manifold Projector** to control coverage and geometric directions via structured perturbations; [Ref1,2] lack explicit uncertainty modeling.
> We will clarify this.
> ## Terminology confusion
> Thank you for pointing out this. To avoid ambiguity, we will revise **"knowledge-guided"** to **“statistics-guided”** throughout the revision.
> ## Theoretical analysis
> Theorems 4.1 and 4.2 are not our main contributions; they formally justify the advantages of HCM. Per Reviewer SPAD's valuable suggestion, we rigorously formalized them while preserving their core intuition. Specifically, Proposition 4.1 aligns with Theorem 4.1 (reduced target scale), and Proposition 4.2 parallels Theorem 4.2 (tighter Jacobian budget for ODE stability). The revisions are as follows:
>
> **Proposition 4.1 (Target Scale).** Let $Y$ be the target, $h:=X_{\mathrm{endo}}$ the endogenous context, and $c$ the covariates. Let $Y_0^H=\mu(h)+\sigma(h)\delta$ denote the HCM source, where $\mu(h)$ is the history-dependent barycenter, $\sigma(h)$ is the history-dependent scale, and $\delta$ is the HCM perturbation. Let $Y_0^G=\xi\sim\mathcal{N}(0,I)$ denote the standard Gaussian source. Define their expected $L_1$ noise magnitudes as $C_H:=\mathbb{E}[\Vert\delta\Vert_1]$ and $C_G:=\mathbb{E}[\Vert\xi\Vert_1]$.
>
> If the conditional residual reduction in centering error $\mathbb{E}[\Vert Y\Vert_1\mid h,c]-\mathbb{E}[\Vert Y-\mu(h)\Vert_1\mid h,c]$ provided by $\mu(h)$ exceeds the combined expected noise magnitudes $\sigma(h)C_H+C_G$,
>
> then HCM yields a smaller conditional $L_1$ matching target than the standard Gaussian source:
>
> $
> \mathbb{E}[\Vert Y-Y_0^H\Vert_1\mid h,c] < \mathbb{E}[\Vert Y-Y_0^G\Vert_1\mid h,c].
> $
>
> **Remark.**
> The assumption is natural in forecasting, where future values typically concentrate around a history-dependent local center rather than the origin of $\mathcal{N}(0,I)$, making the conditional residual reduction more likely to be sufficiently large. Meanwhile, Eq. (7) calibrates $\sigma(h)$, and the normalized manifold perturbation keeps the added noise controlled. Thus, HCM reduces the regression scale, stabilizing gradients and alleviating the empirical optimization burden.
>
> **Proposition 4.2 (Path Regularity).** Let $Y_s^H=sY+(1-s)Y_0^H$ and $Y_s^G=sY+(1-s)Y_0^G$ denote the straight-line matching paths under the HCM source $Y_0^H=\mu(h)+\sigma(h)\delta$ and the standard Gaussian source $Y_0^G=\xi\sim\mathcal{N}(0,I)$, respectively. Define their path-tube radii as $R_H:=\sup_{s\in[0,1]}\Vert Y_s^H-\mu(h)\Vert_1$ and $R_G:=\sup_{s\in[0,1]}\Vert Y_s^G\Vert_1$. Suppose the vector field satisfies a local regularity condition bounding its Jacobian norm by the spatial radius $r$ around any center $m$:
>
> $ \sup_{\substack{s\in[0,1]\\ \Vert y-m\Vert_1\le r}} \Vert \nabla_y v_\theta(y,s,c)\Vert_{1\to1}\le a+br \quad (a\ge0, b>0). $
>
> Under this assumption, a path tube of radius $R$ admits the Jacobian upper bound $a+bR$, which quantifies how strongly the vector field can amplify local perturbations along the path. Accordingly,
> define the corresponding path-local Jacobian budgets as $\mathcal{B}_H:=a+bR_H$ and $\mathcal{B}_G:=a+bR_G$.
>
> If the condition of Proposition 4.1 holds, then HCM yields a smaller expected path-local Jacobian budget than the standard Gaussian source:
> $ \mathbb{E}[\mathcal{B}_H\mid h,c] < \mathbb{E}[\mathcal{B}_G\mid h,c]. $
>
> **Remark.** Proposition 4.2 assumes a local, rather than global, regularity condition. This is reasonable for smooth neural vector fields, since locally bounded Hessians on a fixed tube imply such a bound. Under this assumption, Proposition 4.2 shows that HCM keeps the straight-line matching path in a narrower tube around $\mu(h)$ than the standard source when Proposition 4.1 holds. This yields a smaller path-local Jacobian budget and supports weaker local error amplification under standard ODE stability arguments.
>
>
> **Assumption of Proposition 4.1.** The history-dependent center reduces the conditional residual by more than the total noise budget.
>
> **Assumption of Proposition 4.2.** The vector field is locally regular on the matching-path tube, with Jacobian norm growing at most linearly with the tube radius, and the condition of Proposition 4.1 holds.
> ## Limitations
> We provide the limitations below:
> 1. Unlike single-step deterministic models, KITE requires iterative sampling during inference, inevitably increasing latency.
> 2. The current framework does not support the simultaneous integration of three or more types of statistical priors.

---

> > ### Author Rebuttal · Reviewer_qUpg · 2026-04-02
> >
> > The authors addressed the points I raised properly. I did not see any rejection points raised by other reviewers, so I maintain my recommendation.

---

> > > ### Author Response · Authors · 2026-04-03
> > >
> > > Many thanks to Reviewer qUpg for your positive feedback and for maintaining your recommendation. We are very glad that our responses have properly addressed your points. We deeply appreciate the time and effort you have dedicated to reviewing our work.

---

### Decision · Program_Chairs · 2026-04-30

**Decision:**

Accept (regular)

**Comment:**

The reviewers agree that the paper has a lot of merit: the problem addressed by the paper is relevant, the proposed solution is genuinely novel, the empirical evaluation is broad, the results are strong, and the presentation is clear.
The reviewers pointed to some areas for improvement, which the authors successfully addressed during the rebuttal:
* The theoretical analysis as provided in the paper was not completely rigorous. The authors strenthened their argument during the rebuttal, and committed to incorporating these improvements into the paper.
* The reviewers raised some concerns regarding the experimental protocol, specifically about how baselines are adapted, that ablation studies conducted only on a subset of the data, and how multivariate datasets are treated. The authors addressed these concerns during the rebuttal, and the additional evidence should be incorporated into the paper.